# WatchLog: Efficient and Interpretable Event Reasoning for Endpoint Detection and Response Logs with Multimodal LLMs

Hongyi Zhou [1]   Jianfeng Pan [2]   Min Peng [2]   Shaomang Huang [3]   Xuling Zhang [2]

## Abstract

Endpoint Detection and Response (EDR) systems are crucial for identifying malicious activities on endpoint devices, yet existing methods struggle to efficiently model ultra-long log sequences and to provide interpretable reasoning for security analysts. We propose **WatchLog**, a novel framework that represents raw logs as video-structured data, enabling scalable and expressive video-language modeling of endpoint behaviors. Each event is encoded as a key–value-guided image, and the resulting images are temporally organized into a video sequence. To capture long-range dependencies, WatchLog employs a temporal cross-attention adapter that enables pixel-wise interaction across time. The adapter acts as an auxiliary temporal reasoning pathway, aligning spatial representations with relevant temporal contexts while preserving the original behavioral semantics. We adopt a two-stage pre-training strategy followed by supervised fine-tuning to generate behavior explanations grounded in event-level semantics and detection outcomes. Experiments on our newly constructed EDR8M-20R dataset and a public benchmark demonstrate that WatchLog consistently outperforms state-of-the-art methods in detection accuracy and recall, while offering more interpretable reasoning traces and significantly improved inference efficiency. Extensive ablation studies further support the robustness and interpretability of the proposed method.

## 1. Introduction

As cyberattacks continue to grow in stealth and sophistication (Kaur & Ramkumar, 2022), Endpoint Detection and Response (EDR) systems have become a core component of modern security operations, enabling continuous endpoint monitoring and post-incident forensic analysis (Hassan et al., 2020; Dong et al., 2023; Sharif et al., 2024). Despite the richness of behavioral information contained in EDR logs, their verbosity and complexity introduce substantial challenges. A single log can comprise thousands to tens of thousands of events, each represented by lengthy textual descriptions. Moreover, malicious activities are typically temporally sparse and often intentionally resemble benign behaviors, further complicating reliable identification (Zhou et al., 2026). Beyond accurate detection, practical security workflows also demand interpretable and well-reasoned outputs that can support analysts in making informed decisions.

These challenges highlight the limitations of existing approaches. Most deep learning methods (Bensaoud et al., 2024) lack explicit mechanisms for temporal reasoning and fail to provide interpretability that aligns with the needs of human analysts. Applying LLMs directly (Li et al., 2024; Zhao et al., 2025; Bitaab et al., 2025) to raw EDR logs also encounters significant scalability and efficiency constraints, due to two main factors: (*i*) the ultra-long nature of raw logs often exceeds the practical context windows of current LLMs; and (*ii*) malicious activities are temporally sparse and buried within a multitude of benign events, making stealthy attack patterns difficult to detect.

To overcome these limitations, we propose **WatchLog**, a three-stage framework that progressively converts raw logs into semantic embeddings interpretable by multimodal LLMs, ultimately producing behavioral judgments accompanied by explanatory rationales, as illustrated in Figure 1. Specifically, in the **first stage**, we introduce a novel event-to-image transformation that encodes key-value fields into pixel-level embeddings while preserving field-level semantics. A subsequent image-event alignment pre-training aligns the visual encoder with event-level textual semantics. In the **second stage**, these event-level images are stacked into a video, and a temporal cross-attention block is introduced to capture dependencies across events. This produces

[1]Department of Computer Science and Technology, Tsinghua University. Beijing, China [2]360 Security Technology Inc. Beijing, China [3]China Telecom Cloud Technology Co., Ltd. Beijing, China. Correspondence to: Min Peng <pengmin1@360.cn>.

*Proceedings of the 43rd International Conference on Machine Learning*, Seoul, South Korea. PMLR 306, 2026. Copyright 2026 by the author(s).

compact sequential embeddings that maintain global semantics while alleviating the computational burden of long-context processing. A video-log alignment pre-training further enforces consistency with the holistic semantics of the logs. In the **third stage**, the compact representations are fused with a trainable LLM, fine-tuned under a reasoning-aware objective, enabling WatchLog to generate threat predictions along with interpretable explanations. Representative comparisons against LLM baselines are visualized in the bottom-right radar chart of Figure 1, with detailed numerical results reported in Table 2. We evaluate WatchLog from two perspectives: detection accuracy, measured by exact matches with ground-truth family names, and rationale quality, assessed by advanced LLMs for coherence, consistency, and completeness.

To the best of our knowledge, this is the first work to transform EDR logs into video-structured inputs and leverage a multimodal LLM for reasoning, bridging security log analysis with visual-language modeling. Our main contributions are summarized as follows:

- We introduce WatchLog, the first end-to-end multimodal paradigm for EDR log analysis, offering a spatio-temporal encoding perspective that unifies visual and textual modeling for precise behavioral judgments with interpretable rationale generation.

- We design a three-stage training pipeline, comprising image–event and video–log pre-training to effectively align visual and textual semantics, followed by supervised fine-tuning, enabling robust adaptation to downstream EDR reasoning tasks.

- We construct a large-scale, expert-annotated EDR corpus of 8 million events, covering diverse behaviors with detailed analytic rationales. This dataset fills a critical gap and serves as a benchmark for advancing endpoint security research.

- Extensive experiments show that WatchLog consistently outperforms state-of-the-art methods in threat detection and rationale generation, while substantially reducing GPU memory usage and inference latency compared to standard LLM models, confirming its feasibility for industrial deployment.

## 2. Related Work

This section reviews relevant EDR solutions and associated technologies, providing valuable insights for our research.

### 2.1. Machine Learning Methods

Early endpoint threat detection systems primarily relied on rule-based signatures (Hassan et al., 2020), which exhibited limited adaptability to previously unseen or evolving attacks. Machine learning (ML) approaches partially addressed these limitations by extracting hand-crafted features from logs or files and training supervised classifiers (Najafi et al., 2024). For instance, WATSON (Zeng et al., 2021) leveraged inverse document frequency and hierarchical clustering to construct semantic representations of events, while (Kumar et al., 2022) explored image-based texture features in combination with Naïve Bayes classifiers. Despite these advances, such methods remain heavily dependent on domain-specific feature engineering. Moreover, their static feature representations struggle to capture the evolving semantics of event sequences, which ultimately limits their effectiveness against sophisticated and stealthy threats.

### 2.2. Deep Learning Methods

Deep learning (DL) enables automatic feature extraction, allowing models to learn attack patterns directly from raw logs (Yu et al., 2019). Early studies mainly adopted recurrent architectures (Ring et al., 2021) to capture sequential dependencies. More recent graph-based methods, such as ProGraPher (Yang et al., 2023), leverage graph representations to model structural relationships and combine them with sequence models for improved detection. Other studies explore adversarial debiasing (Tsai et al., 2024) or self-supervised pre-training (Sharif et al., 2024) to enhance robustness and generalization. Despite these advances, most DL-based approaches still treat logs as either lengthy text or simple event sequences, thus struggling to capture fine-grained semantics and to model the temporal evolution of the log sequences. These limitations motivate our spatio-temporal modeling approach for EDR log analysis.

### 2.3. LLM-based Approaches

Large Language Models (LLMs) (Brown et al., 2020) have recently shown strong potential across security tasks such as malware analysis, phishing detection, and intrusion prevention (Li et al., 2024; Hu et al., 2024; Zhou et al., 2025). Compared with conventional DL approaches, LLMs exhibit superior capabilities in semantic understanding and logical reasoning (Snell et al., 2025), making them particularly promising for identifying malicious behaviors in log data. For instance, Mal-LLM (Xue et al., 2024) leveraged chain-of-thought prompting to guide LLMs in capturing the semantic intent of malicious code, combined with classical ML classifiers. AppPoet (Zhao et al., 2025) employed LLMs to generate rich linguistic descriptions of software artifacts, improving both detection performance and interpretability.

Nevertheless, directly applying LLMs to raw EDR logs remains challenging. Log sequences can easily exceed hundreds of thousands of tokens, which severely strains context retention and inference efficiency. Moreover, malicious

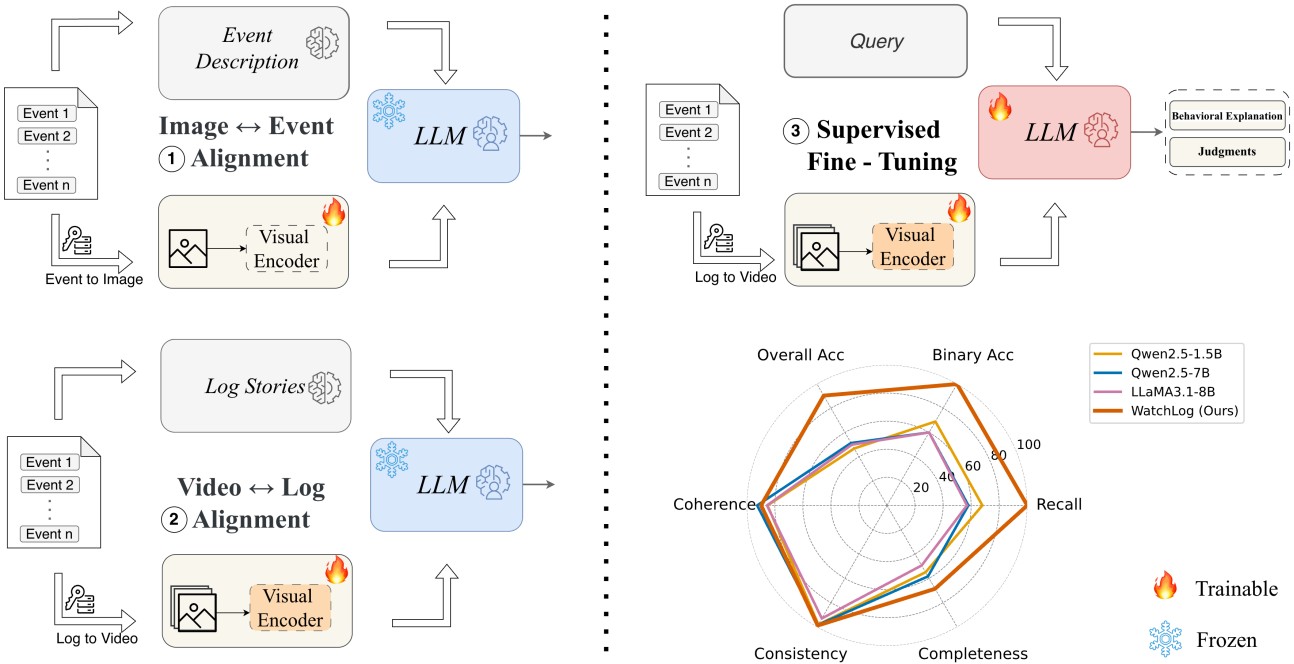

*Figure 1.* Overview of WatchLog Framework with Representative Results

events are typically sparse and often closely resemble benign operations (Sim & Borthwick, 2018; Dong et al., 2023; Zhou et al., 2026), further complicating reliable reasoning over long contexts. In contrast, we propose a novel multimodal framework that more effectively harnesses the semantic understanding and reasoning capabilities of LLMs to detect and interpret malicious behavior.

### 2.4. Long-Context Understanding in LLMs

Some recent studies extend the context length of LLMs, including RoPE-based extrapolation (YaRN) (Peng et al., 2024), bi-level attention mechanisms (SelfExtend) (Jin et al., 2024), and progressive interpolation strategies (LongRoPE) (Ding et al., 2024). While these approaches can expand the context window to millions of tokens, they remain computationally expensive and are poorly suited to the ultra-long, high-volume sequences characteristic of EDR data. In contrast, our approach transforms logs into structured image- and video-like representations for multimodal LLM processing, improving computational efficiency while preserving fine-grained event semantics and long-range temporal dynamics. This design also offers a more practical paradigm for reasoning-aware malicious behavior detection.

### 2.5. Video-Text Representation Learning

Recent advances in video-text representation learning have improved the modeling of long temporal sequences and

multimodal reasoning. VideoCoCa (Yan et al., 2022) and VideoPrism (Zhao et al., 2024) extend large-scale video-language pretraining via temporal aggregation and multimodal supervision. More recent models, such as Intern-Video2 (Wang et al., 2024) and LLaVA-Video (Zhang et al., 2025), further strengthen temporal modeling and integrate video representations with LLM-based reasoning for long-video understanding. These methods inspire our design for modeling long sequential observations. Specifically, EDR logs can be viewed as spatio-temporal structures, where event fields form spatial components and event sequences capture temporal dynamics. We adapt multimodal modeling and temporal compression to security logs, enabling efficient reasoning over long behavioral traces while preserving fine-grained attack semantics.

## 3. Methods

We propose a three-stage framework for malicious behavior detection and reasoning on EDR logs. The core idea is to progressively transform raw logs into structured spatio-temporal representations interpretable by multimodal language models, addressing challenges such as ultra-long sequences, noisy events, and the need for interpretable reasoning. In the first stage, individual events are converted into images and aligned with their textual descriptions; in the second stage, these event-level images are organized into a video-structured sequence and aligned with log-level

descriptions; in the third stage, raw logs are converted into structured representations and fine-tuned to generate behavioral judgments and explanatory rationales. This progressive design captures both event-level semantics and temporal dynamics, while leveraging multimodal reasoning to achieve precise understanding of log behaviors. Appendix I provides a detailed analysis of the mathematical properties and computational complexity of each stage of our method.

### 3.1. Image ↔ Event Alignment

In this stage, our goal is to construct structured event-level representations that support both image-event alignment and downstream modeling. Given a raw event $E$, we process it through two parallel pathways: (i) a *textual pathway*, where the event is reformulated by GPT (Hurst et al., 2024) into detailed natural language descriptions $\mathcal{D}$ (see Appendix Table 8 for prompt design); and (ii) a *visual pathway*, where the event is decomposed into key-value pairs $\{\mathbf{x}_m = (k_m, v_m)\}_{m=1}^{M}$, and each pair $\mathbf{x}_m$ is tokenized into embeddings $\{e_{m1}, \ldots, e_{mL_m}\}$ with $e_{m\ell} \in \mathbb{R}^d$ and $L_m$ denoting its token length. To accommodate variable-length pairs, we introduce a *kvEmbedding* layer that applies a trainable projection with softmax-based weighting, yielding a fixed-dimensional embedding:

$$\tilde{e}_m = \sum_{\ell=1}^{L_m} \frac{\exp(W^\top e_{m\ell})}{\sum_{\ell=1}^{L_m} \exp(W^\top e_{m\ell})} \cdot e_{m\ell}, \qquad (1)$$

with $W \in \mathbb{R}^d$, $d$ denoting the hidden dimension of the token embeddings. The resulting embeddings $\{\tilde{e}_1, \ldots, \tilde{e}_M\}$ are then projected to the ViT hidden dimension $c$ and reshaped into an image-like tensor $\tilde{E} = \text{Reshape}([\tilde{e}_1; \ldots; \tilde{e}_M]) \in \mathbb{R}^{c \times h \times w}$, with $M = hw$.

To align the two paths, we perform image-to-description pre-training, where each event $E$ is processed to obtain an image $\tilde{E}$, which is aligned with its GPT-generated textual description $\mathcal{D}$.

$$\mathcal{J}_{IEA} = -\sum_{l=1}^{L_I} \log P_{\theta'}(\mathcal{D}_l \mid \mathcal{D}_{<l}, E), \qquad (2)$$

where $L_I$ denotes the number of tokens for the textual description $\mathcal{D}$, $\theta'$ denotes all trainable parameters in this stage. This alignment grounds the visual representations in human-interpretable semantics, thereby enhancing their utility for subsequent log-level modeling and multimodal reasoning.

### 3.2. Video ↔ Log Alignment

In this stage, we extend the framework to temporal semantic alignment of logs. Given a log file $\mathcal{L}$ with $T$ events, we also process it through two parallel pathways: (i) a *textual pathway*, where the log sequence is reformulated by GPT (Hurst et al., 2024) into detailed natural language descriptions (see Appendix Table 9 for prompt design); and (ii) a *visual pathway*, where the event-level images obtained from the previous stage are chronologically stacked into a video-like tensor $\mathcal{E} = [\tilde{E}_1, \tilde{E}_2, \ldots, \tilde{E}_T] \in \mathbb{R}^{T \times c \times h \times w}$. To reduce redundancy while preserving salient temporal information, we introduce a temporal cross-attention block. Specifically, we first apply adaptive average pooling to generate a compact query set $Q = \mathcal{E}_{pool} \in \mathbb{R}^{t \times c \times h \times w}$ ($t \ll T$), while the original video-like embeddings serve as keys ($K$) and values ($V$). The cross-attention is then applied at each pixel location:

$$\tilde{\mathcal{E}}_{i,j} = \text{Softmax}\left(\frac{Q_{i,j} W_Q (K_{i,j} W_K)^\top}{\sqrt{c}}\right)(V_{i,j} W_V), \quad (3)$$

where $K, V = \mathcal{E}$, $W_Q, W_K, W_V \in \mathbb{R}^{c \times c}$ are trainable projections, $\tilde{\mathcal{E}}_{i,j} \in \mathbb{R}^{t \times c}$, and $1 \le i \le h, 1 \le j \le w$ denote pixel coordinates on the image grid. This operation condenses the $T$ frames into $t$ abstracted frames while maintaining spatial consistency, yielding a video-level representation that highlights discriminative temporal dynamics.

To align the two modalities, we perform video-log pre-training, where each log $\mathcal{L}$ is processed to obtain the abstracted video $\tilde{\mathcal{E}}$, which is aligned with the GPT-generated log stories $\mathcal{S}$. We adopt an autoregressive objective: the multimodal model takes the original log $\mathcal{L}$ as input and generates the description tokens $\{\mathcal{S}_1, \ldots, \mathcal{S}_{L_V}\}$, optimized by the negative log-likelihood loss.

$$\mathcal{J}_{VLA} = -\sum_{l=1}^{L_V} \log P_{\theta''}(\mathcal{S}_l \mid \mathcal{S}_{<l}, \mathcal{L}), \qquad (4)$$

where $L_V$ denotes the number of tokens for the log stories $\mathcal{S}$, $\theta''$ denotes all trainable parameters in this stage.

### 3.3. Supervised Fine-Tuning

Moving from pre-training to task-specific optimization, the model is trained to produce both categorical predictions and human-interpretable rationales. The inputs consist of (i) raw logs $\mathcal{L}$ and (ii) an external query text $q$. The raw logs are processed through the visual encoder from the video-log alignment stage to obtain $\tilde{\mathcal{E}}$, which is then fine-tuned end-to-end with the LLM under supervised objectives.

Specifically, each training instance is paired with a ground-truth family label $y$, and a natural-language rationale $r$ (see Appendix Table 10 for details) that highlights the evidence of malicious behavior. These are concatenated into a unified autoregressive target sequence $\mathbf{y} = \{y_1, \ldots, y_L\}$, with the family token first, followed by reasoning tokens. The model is trained by minimizing the negative log-likelihood:

$$\mathcal{J}_{SFT} = -\sum_{l=1}^{L} \log P_{\theta^*}((y, r)_l \mid (y, r)_{<l}, (\mathcal{L}, q)), \quad (5)$$

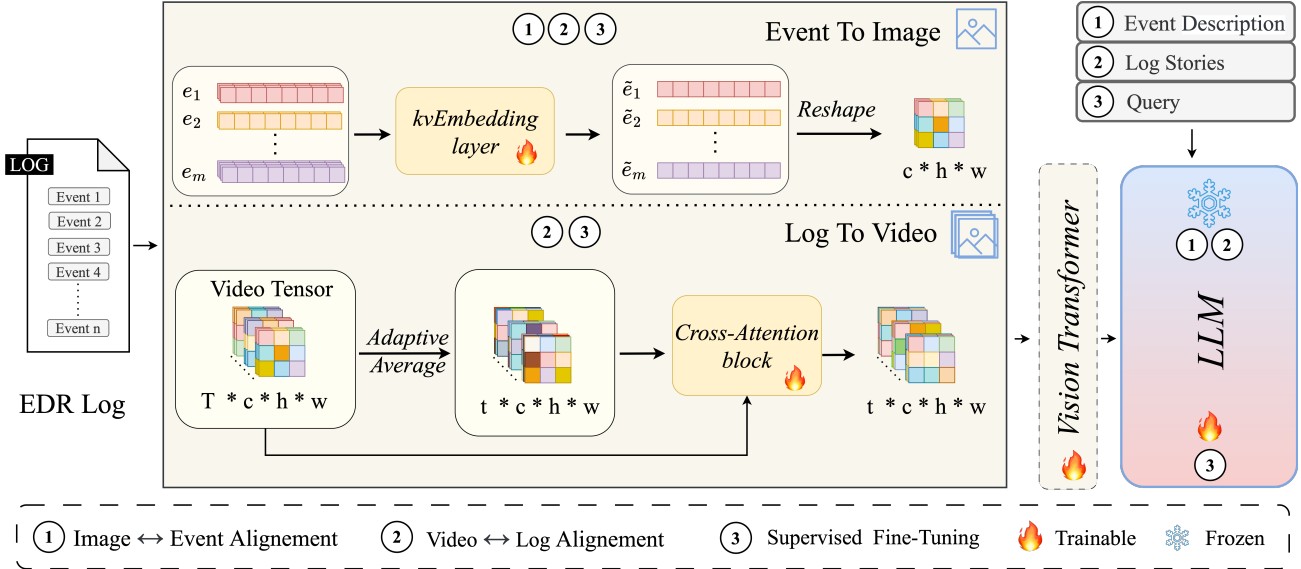

*Figure 2.* Overview of the training process for each stage of the **WatchLog** framework. In Stage 1, events are parsed and transformed into pixel embeddings via the kvEmbedding layer (MLP + softmax aggregation), and aligned with their textual descriptions through image-event pre-training. In Stage 2, images are temporally stacked and processed by an additional cross-attention block to produce compact video-like representations, which are further aligned with overall log semantics via video-log pre-training. In Stage 3, the unfrozen language model backbone is fine-tuned end-to-end together with the visual encoder to generate behavioral predictions and explanatory rationales. All stages share and fine-tune the same Vision Transformer (ViT) structure.

where $\theta^*$ denotes all trainable parameters of the model. $L$ represents the total number of tokens in the ground-truth family name and its corresponding rationale. Overall, WatchLog introduces a three-stage training strategy that progressively decouples representation alignment, temporal modeling, and reasoning, enabling scalable and interpretable analysis of large-scale security logs.

## 4. Experiments

### 4.1. Experimental Setup

**Dataset.** We introduce EDR8M-20R, the first large-scale, well-annotated benchmark dataset designed for end-to-end evaluation of EDR systems, and conduct extensive experiments on this dataset. Compared with existing benchmarks (Man Duc Trong et al., 2020; Zengy et al., 2022; Yang et al., 2023; Sharif et al., 2024), EDR8M-20R is publicly released and uniquely combines complete real-world endpoint execution traces with expert-verified reasoning paths and calibrated threat family names. The data collection and construction process is described in Appendix D. We also provide details on dataset release and access in Appendix E.

EDR8M-20R covers 20 behavior families (including benign behavior), with 100 samples per family. We randomly select 80 samples from each family for training and reserve the remaining 20 for testing. To further assess generalization, we construct two additional evaluation sets: an independent

test set comprising 1,000 additional samples from the same 20 families, and an out-of-distribution (OOD) test set containing 112 samples from previously unseen families. To support supervised fine-tuning (SFT), we generate explanatory rationales for each sample using an LLM (Hurst et al., 2024) with human verification. For OOD samples, the rationales indicate only that the behavior is malicious, without revealing specific family labels to the model. Details of rationale generation are described in Appendix C.

**Baselines.** We compare against three representative traditional methods: LGTF (Kumar et al., 2022), ADE (Tsai et al., 2024), and DrSec (Sharif et al., 2024); two general-purpose LLM baselines under zero-shot testing: GPT-5 (Singh et al., 2025), Gemini-2.5-Pro (Comanici et al., 2025), DeepSeek-R1-671B (Guo et al., 2025) and Qwen3-235B-A22B (Yang et al., 2025); and task-specific fine-tuned methods (Bitaab et al., 2025; Zhou et al., 2025) based on LLMs (e.g., Qwen2.5-1.5/7B-Instruct (Yang et al., 2024) and LLaMA3.1-8B-Instruct (Grattafiori et al., 2024)). Further implementation details are provided in Appendix F.

**Training Details.** WatchLog is built on the Qwen2-VL-2B backbone by default, with the effect of different backbone sizes evaluated in subsequent ablation studies. For image-event alignment, we replace the *pathEmbedding* layer in Qwen2-VL with a *kvEmbedding* layer, where the number of key–value pairs $M$ is determined statistically from the training set and set to 144. For video-log alignment, a cross-

*Table 1.* Pseudocode of WatchLog under the proposed three-stage framework.

---

**Algorithm 1: Algorithm of the WatchLog Framework**

---

**Input:** raw log file $\mathcal{L}$ with $T$ events; Query $q$.
**Output:** attack family name $y$ and rationale $r$.

**Stage 1: Image-Event Alignment**
    **for** event $E_i$ in $\mathcal{L}$ **do**
        decompose $E_i \rightarrow \{\mathbf{x}_m = (k_m, v_m)\}_{m=1}^{M}$
        tokenize $\mathbf{x}_m \rightarrow \{e_{m\ell}\}_{\ell=1}^{L_m}$
        transform $\text{kvEmbedding}(e_m) \rightarrow \tilde{e}_m$
        reshape $\text{Reshape}([\tilde{e}_1; \ldots; \tilde{e}_M]) \rightarrow \tilde{E}_i$
        $\mathcal{J}_{IEA} = -\sum_{l=1}^{L_I} \log P_{\theta'}(\mathcal{D}_l \mid \mathcal{D}_{<l}, E_i)$
    **end for**

**Stage 2: Video-Log Alignment**
    $\mathcal{L} \rightarrow \mathcal{E} \in \mathbb{R}^{T \times c \times h \times w}$
    $\text{AdaptiveAvgPool}(\mathcal{E}) \rightarrow \mathcal{E}_{pool}$
    set $Q = \mathcal{E}_{pool}, K = \mathcal{E}, V = \mathcal{E}$
    $\text{CrossAttn}(Q, K, V) \rightarrow \tilde{\mathcal{E}}$
    $\mathcal{J}_{VLA} = -\sum_{l=1}^{L_V} \log P_{\theta''}(S_l \mid S_{<l}, \mathcal{L})$

**Stage 3: Supervised Fine-Tuning**
    feed $(\mathcal{L}, q) \rightarrow \tilde{\mathcal{E}}$
    generate $y$ and $r$
    $\mathcal{J}_{SFT} = -\sum_{l=1}^{L} \log P_{\theta*}((y, r)_l \mid (y, r)_{<l}, (\mathcal{L}, q))$

---

attention block is inserted after the *kvEmbedding* layer, with a temporal compression dimension $t$ set to 64 by default. Training is performed with a batch size of 16 over 10 epochs using the AdamW optimizer with an initial learning rate of $1 \times 10^{-5}$, together with a cosine annealing scheduler. All experiments are conducted on a single node equipped with eight NVIDIA H100 (80GB) GPUs.

**Metrics.** For binary detection, we report *Binary Accuracy*, *Recall*, and *False Alarm*, which measure the overall correctness in distinguishing benign and malicious samples, the ability to detect malicious behaviors, and the proportion of benign samples misclassified as malicious, respectively. For multi-family behavior detection, we adopt *Overall Accuracy*, defined as the proportion of behavior families correctly identified, where a prediction is considered correct only if the generated family name exactly matches the ground truth. For rationale evaluation, explanations are assessed in terms of *Coherence*, *Consistency*, and *Completeness*. To mitigate evaluation bias, we average scores independently produced by GPT-5 (Singh et al., 2025), Gemini-2.5-Pro (Comanici et al., 2025), and Qwen3-235B-A22B (Yang et al., 2025), and normalize the results to a 100-point scale. Evaluation prompts are provided in Appendix G. For trainable methods, the reported results correspond to the best validation performance achieved during training.

## 4.2. Comprehensive Evaluation

We evaluate WatchLog against traditional EDR detection systems, general-purpose large-scale LLMs, and current LLM-based methods fine-tuned for downstream tasks. As shown in Table 2, WatchLog achieves the best overall performance in terms of detection accuracy (Binary Accuracy 99.8%, Recall 100%, False Alarm 5.0%) and superior reasoning quality (Rationale Completeness 68.5), establishing a new state of the art on EDR8M-20R; mean and standard deviation over repeated runs are reported in Appendix H.

**Detection Performance.** Traditional methods such as LGTF and ADE achieve high recall ($\geq 99\%$) but suffer from extremely high false alarm rates (65–100%), resulting in poor overall accuracy ($\leq 48.8\%$). Zero-shot LLMs (GPT-5, Gemini-2.5-Pro, DeepSeek-R1-671B, and Qwen3-235B-A22B) achieve relatively balanced performance but remain unstable, with overall accuracy around 10%. Instruction-tuned LLMs reduce the false alarm rate (near 0%) but underfit long EDR sequences, yielding binary accuracy around 60–69% and overall accuracy below 51.3%. In contrast, WatchLog achieves both high recall and low false alarm rates, substantially outperforming all baselines in overall accuracy.

**Reasoning Quality.** In addition to detection, we evaluate generated rationales using three metrics: *Coherence*, *Consistency*, and *Completeness*. As shown in Table 2, WatchLog achieves the highest *Completeness* score (68.5), substantially outperforming all LLM baselines. While competing models demonstrate relatively strong *Coherence* and *Consistency* (e.g., Qwen2.5-1.5B-Instruct and Qwen2.5-7B-Instruct), they often omit key causal chains—the most critical component captured by the *Completeness* metric. These results indicate that our implicit log-to-video representation provides stronger semantic grounding, enabling WatchLog to both detect malicious activities and generate interpretable explanations essential for practical EDR applications.

## 4.3. Generalization Evaluation

Given the adversarial nature of cybersecurity, where attackers continually adapt existing attack techniques or create new variants to evade detection, evaluating model generalization is crucial for practical EDR systems.

**Evaluation on the independent and out-of-distribution test sets.** As described in the dataset section, we constructed additional independent test sets and out-of-distribution (OOD) attack sets to evaluate the robustness of the model. As shown in Table 3, while baseline models achieve moderate performance on the independent test set, their recall drops sharply on the OOD test set (17.0–32.0%), indicating limited adaptability to novel threats. In contrast, WatchLog maintains near-perfect performance on the independent test

*Table 2.* Comparison of detection performance (%) and reasoning quality on EDR8M-20R. $\diamond$ indicates zero-shot evaluation. $\spadesuit$ denotes supervised fine-tuning (SFT). Reasoning metrics are only reported for models capable of generating rationales. BA = Binary Accuracy, R = Recall, FA = False Alarm, and OA = Overall Accuracy. Cohe. = Coherence, Cons. = Consistency, and Comp. = Completeness.

| Method | Detection performance | | | | Reasoning Quality | | |
|---|---|---|---|---|---|---|---|
| | BA ↑ | R ↑ | FA ↓ | OA ↑ | Cohe. ↑ | Cons. ↑ | Comp. ↑ |
| LGTF (Kumar et al., 2022) | 92.8 | 96.0 | 65.0 | 48.8 | – | – | – |
| ADE (Tsai et al., 2024) | 93.8 | 99.0 | 100.0 | 20.5 | – | – | – |
| DrSec (Sharif et al., 2024) | 92.0 | 96.0 | 90.0 | 38.8 | – | – | – |
| GPT-5$^\diamond$ (Singh et al., 2025) | 63.8 | 62.4 | 10.0 | 13.5 | – | – | – |
| Gemini-2.5-Pro$^\diamond$ (Comanici et al., 2025) | 76.3 | 77.6 | 50.0 | 9.0 | – | – | – |
| DeepSeek-R1-671B $^\diamond$ (Guo et al., 2025) | 87.3 | 90.0 | 70.0 | 8.8 | – | – | – |
| Qwen3-235B-A22B$^\diamond$ (Yang et al., 2025) | 93.3 | 97.0 | 85.0 | 7.8 | – | – | – |
| Qwen2.5-1.5B-Instruct$^\spadesuit$ (Yang et al., 2024) | 69.0 | 68.0 | 15.0 | 46.8 | 94.4 | 97.8 | 55.4 |
| Qwen2.5-7B-Instruct$^\spadesuit$ (Yang et al., 2024) | 60.0 | 58.0 | **0.0** | 51.3 | **96.4** | **98.2** | 59.2 |
| LLaMA3.1-8B-Instruct$^\spadesuit$ (Grattafiori et al., 2024) | 60.0 | 57.0 | **0.0** | 50.0 | 91.3 | 92.9 | 50.8 |
| **WatchLog (Ours)** | **99.8** | **100.0** | 5.0 | **90.3** | 93.6 | 95.0 | **68.5** |

*Table 3.* Performance on the independent test set and out-of-distribution test set.

| Method | Independent Test Set | | | | Out-of-Distribution Test Set | |
|---|---|---|---|---|---|---|
| | FA↓ | R↑ | BA↑ | OA↑ | R↑ | Comp.↑ |
| Qwen2.5-1.5B-Instruct | 24.0 | 73.0 | 72.7 | 45.5 | 32.0 | 39.6 |
| Qwen2.5-7B-Instruct | **0.0** | 58.0 | 60.4 | 52.8 | 21.0 | 35.6 |
| LLaMA3.1-8B-Instruct | 2.0 | 61.0 | 62.6 | 51.9 | 17.0 | 39.0 |
| **WatchLog (Ours)** | 4.0 | **100.0** | **99.6** | **88.2** | **74.0** | **40.2** |

*Table 4.* Evaluation results on ATLASv2, which is a binary classification task (benign vs. malicious).

| Method | FA ↓ | R ↑ | BA ↑ |
|---|---|---|---|
| Qwen2.5-1.5B-Instruct | 100.0 | **100.0** | 44.4 |
| Qwen2.5-7B-Instruct | **5.0** | 13.0 | 58.3 |
| LLaMA3.1-8B-Instruct | 85.0 | 75.0 | 41.7 |
| **WatchLog (Ours)** | 50.0 | 88.0 | **66.7** |

set (100.0% recall, 99.6% binary accuracy, and only 4.0% false alarms) while also achieving 74.0% recall and the highest *Completeness* score on OOD attacks. These results demonstrate that WatchLog not only ensures reliable recognition under matched conditions but also generalizes effectively to previously unseen attacks, providing the robustness and transferability required for real-world deployment.

**Evaluation on the ATLASv2 Dataset.** To assess robustness beyond our curated dataset, we further evaluate our model on the external ATLASv2 benchmark in a zero-shot setting. We restrict our evaluation to the Carbon Black Cloud subset, as it is the only component of ATLASv2 that reflects realistic EDR telemetry; other subsets (e.g., Firefox browsing logs or Wireshark DNS traces) deviate

substantially from practical EDR scenarios. As shown in Table 4, Qwen2.5-1.5B-Instruct achieves perfect recall by aggressively labeling logs as malicious, leading to substantial over-prediction, whereas Qwen2.5-7B-Instruct adopts a more conservative strategy that reduces false alarms but misses the majority of attack instances. LLaMA3.1-8B-Instruct lies between these two extremes, yet its overall accuracy remains limited. In contrast, WatchLog delivers a more balanced performance, achieving 88% recall and the highest binary accuracy (66.7%), which suggests stronger generalization under distribution shift.

### 4.4. Computational Efficiency Analysis

Table 5 reports GPU memory usage (GPU-MU) and time-to-first-token (TTFT) across increasing input lengths (64K, 128K, 256K, 512K, and 1024K tokens), providing a quantitative assessment of computational behavior under long-context settings. For baseline LLMs, including Qwen2.5-7B-Instruct and LLaMA3.1-8B-Instruct, both GPU-MU and TTFT grow nearly linearly or super-linearly with sequence length, indicating rapidly increasing resource demands as the context scales. In contrast, our model exhibits substantially more stable scaling: GPU-MU increases from 11.27GB to 48.61GB, while TTFT grows from 0.63s to

*Table 5.* Comparison of GPU memory usage (GPU-MU, in GB) and inference latency (TTFT, in seconds) under different input lengths.

| | GPU-MU (GB) | | | | | TTFT (s) | | | | |
|---|---|---|---|---|---|---|---|---|---|---|
| | 64K | 128K | 256K | 512K | 1024K | 64K | 128K | 256K | 512K | 1024K |
| Qwen2.5-1.5B-Instruct | **9.03** | 15.09 | 26.87 | 50.77 | 98.46 | 1.60 | 5.51 | 20.49 | 78.81 | 259.30 |
| Qwen2.5-7B-Instruct | 27.13 | 39.67 | 64.74 | 114.88 | 215.17 | 4.33 | 13.99 | 50.23 | 151.76 | 486.19 |
| LLaMA3.1-8B-Instruct | 31.02 | 46.69 | 78.03 | 140.71 | 266.07 | 5.31 | 17.61 | 53.73 | 167.43 | 521.72 |
| **WatchLog (Ours)** | 11.27 | **12.67** | **15.47** | **26.81** | **48.61** | **0.63** | **0.66** | **0.69** | **0.79** | **1.11** |

1.11s as the input length extends to one million tokens. Relative to the smaller Qwen2.5-1.5B-Instruct model, our approach reduces GPU-MU by 50.6% and improves TTFT by a factor of 233.6×. This behavior is attributed to the architecture of WatchLog, where kvEmbedding and selective KV aggregation constrain the effective sequence length involved in downstream reasoning. To further demonstrate deployment feasibility, we evaluate the models in Table 5 on NVIDIA H100 80GB GPUs. Under the 1024K-token context setting, WatchLog runs on a single GPU, whereas Qwen2.5-7B-Instruct requires at least three GPUs under the same configuration. This demonstrates improved deployment scalability and practical applicability of our approach. Overall, these measurements indicate that WatchLog can process million-token EDR logs with bounded memory and latency, demonstrating clear feasibility for industry-level deployment.

### 4.5. Ablation Study

We conduct ablation experiments to systematically analyze the contribution and effectiveness of each component in WatchLog. Further experimental results and illustrative examples can be found in Appendix K.

**Effects of Different Alignment Pre-training.** Table 6 summarizes the effects of different alignment pre-training strategies, including no alignment pre-training (w/o all), image-level alignment only (+ $I_{alig.}$), and the default setting with both image- and video-level alignment (+ $V_{alig.}$). Models trained without alignment pre-training show consistent performance degradation. Adding image–event alignment yields a noticeable improvement, and further incorporating video–log alignment achieves the best overall performance. Together, these results are consistent with image- and video-level alignment providing complementary benefits.

**Effects of Varying the Temporal Value $t$.** The temporal value $t$ determines the degree of temporal abstraction. As shown in Figure 3 (a), smaller values of $t$ correspond to higher temporal compression and lower computational cost, while larger values retain more information at the expense of increased computational overhead. In our experiments, $t = 64$ provides a moderate level of temporal abstraction, preserving key temporal cues while mitigating noise from

*Table 6.* Ablation study on alignment pre-training.

| Method | FA ↓ | OA ↑ | Cohe. ↑ | Cons. ↑ | Comp. ↑ |
|---|---|---|---|---|---|
| w/o all | 100.0 | 5.3 | 94.5 | 94.2 | 32.1 |
| + $I_{alig.}$ | 5.0 | 88.8 | 92.1 | 93.6 | 67.2 |
| + $V_{alig.}$ | 5.0 | 90.3 | 93.6 | 95.0 | 68.5 |

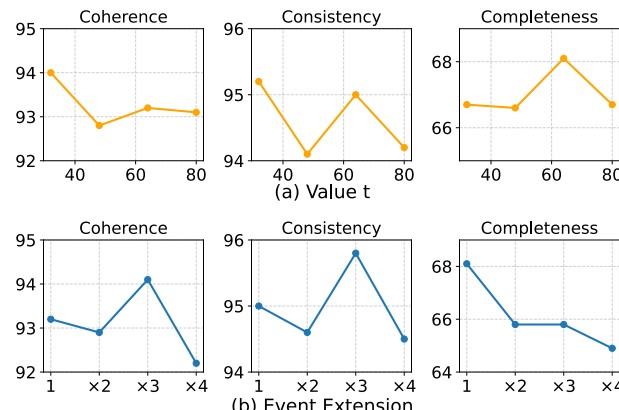

*Figure 3.* Ablation studies on temporal value $t$ and event extension.

long event sequences. The influence of different $t$ values on model performance is quantified in Appendix Tables 13, which correspond to the results illustrated in Figure 3 (a).

**Robustness to Event Sequence Extension.** We evaluate the method on ultra-long log sequences by extending event sequences by factors of 2, 3, and 4 to assess robustness with respect to event number scaling. As shown in Figure 3 (b), performance decreases only marginally despite the substantial increase in sequence length. This demonstrates that our log-to-video representation and temporal aggregator are highly effective at preserving robustness and scalability under extreme temporal horizons. The results underline the framework's ability to handle long sequences without significant performance loss. Numerical results corresponding to Figure 3 (b) are provided in Appendix Tables 14.

**VLM Backbone Scaling.** To evaluate the impact of the underlying VLM backbone on WatchLog, we replace the default model with larger variants, including Qwen2.5-VL-3B-

*Table 7.* Ablation study on VLM backbone scaling.

| Method | FA ↓ | OA ↑ | Cohe. ↑ | Cons. ↑ | Comp. ↑ |
|---|---|---|---|---|---|
| WatchLog-2B | 5.0 | 90.3 | 93.6 | 95.0 | 68.5 |
| WatchLog-3B | 0.0 | 91.3 | 95.8 | 96.6 | 74.4 |
| WatchLog-7B | 0.0 | 92.5 | 96.3 | 97.3 | 78.2 |

Instruct and Qwen2.5-VL-7B-Instruct (Yang et al., 2024), and perform evaluations under identical settings. As reported in Table 7, increasing backbone size leads to consistent improvements across key metrics. Moving from 2B to 3B reduces false alarms from 5.0% to 0.0% and yields increases in overall accuracy and rationale metrics. Scaling further to the 7B backbone provides additional gains, reaching an overall detection accuracy of 92.5% and reasoning quality scores of 96.3, 97.3, and 78.2 in *Coherence*, *Consistency*, and *Completeness*, respectively. These results indicate that WatchLog's performance scales with the capacity of the underlying VLM, and that larger backbones can be incorporated without changes to the model architecture.

## 5. Conclusion

In this paper, we present WatchLog, a framework that reformulates endpoint security logs into video–language representations for malicious behavior detection. Experimental results demonstrate that WatchLog consistently outperforms baseline methods while producing high-quality, interpretable outputs and generalizes strongly to unseen attacks on both curated out-of-distribution test sets and public benchmarks. Efficiency evaluations further show that WatchLog can handle ultra-long log sequences with bounded memory and latency, enabling practical deployment. Future work will explore incorporating richer cybersecurity knowledge and integrating WatchLog into broader security systems to enable more adaptive defenses.

## Impact Statement

The primary positive societal impact of this work lies in strengthening cybersecurity defenses. Improved detection and interpretability can help organizations respond to cyber threats more effectively and assist security analysts in decision-making. The emphasis on efficiency and scalability also enhances the feasibility of deploying these techniques in real-world environments with ultra-long log data. At the same time, care must be taken to manage potential risks, such as false positives or adversarial evasion of detection systems, and to ensure that the method is used in appropriate security contexts. Overall, this work introduces a multimodal modeling paradigm for long-sequence reasoning in security logs, with an expected positive impact when applied responsibly.

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

## A. Prompt for Generating Event Descriptions

*Table 8.* Prompts used for event descriptions generation.

---

You are a professional cybersecurity analyst. Given the following EDR log (a single event), generate a professional text description.

###Requirements:

1. Base the description only on the explicit information in the log. Do not invent or speculate.

2. Use clear, concise, full sentences. Avoid redundancy.

3. Cover all key elements present in the log, such as timestamp, host, process, command line, file activity, network connections, and hash values.

4. Output should be a single natural-language paragraph, not a list or JSON.

### Input
EDR log event: [*event*]

---

## B. Prompt for Generating Log Stories

*Table 9.* Prompts used for log stories generation.

---

You are a professional cybersecurity analyst. You are given a log consisting of multiple paragraphs, where each paragraph represents a text report recorded at different time points. Your task is to generate a comprehensive yet concise summary of the entire log.

###Requirements:

1. Completeness: Cover all major points and important details from the entire log without omitting key information.

2. Conciseness: Ensure the summary remains clear, well-structured, and free of redundancy, while still preserving important details.

3. Coherence: Present the summary in a logically organized and fluent manner, not as a fragmented list of sentences.

4. Fidelity: Do not introduce assumptions, interpretations, or information that is not present in the original log.

### Input
LOG: [*log*]

---

## C. Behavioral Rationale Construction

To enhance the interpretability of EDR detection results during the SFT stage, we construct behavioral rationale data that explicitly explain why a given log is associated with a specific attack or activity family. For each log, a structured prompt (shown in Table 10) is designed to guide GPT (Hurst et al., 2024) in generating these rationales. Each prompt consists of three components: (i) descriptions of the individual events, (ii) the target behavioral family, and (iii) a brief description of the family's characteristics.

To ensure the accuracy and reliability of the generated rationales, all outputs were reviewed and verified by human security experts. Experts checked the consistency of the rationale with the underlying log, corrected errors, and refined the explanations to ensure that they accurately reflect the behavioral patterns associated with each family. This human-in-the-loop verification process guarantees high-quality, trustworthy rationale data for subsequent model training and evaluation.

## D. Data Collection

The availability of large-scale and high-quality datasets, particularly those encompassing diverse attack types and complex behavioral patterns, is essential for validating algorithms and enhancing the detection capabilities of EDR systems. Endpoint Detection and Response (EDR) logs record fine-grained system activities generated by endpoints such as user machines and servers. Each log entry typically corresponds to an atomic system event, including process executions, file operations, network connections, and registry modifications, forming a continuous trace of system behavior over time. These logs are

*Table 10.* Prompts used for rationale data generation.

---

**Prompt for a Specified Behavioral Family**

---

You are a professional cybersecurity analyst. You are given a log consisting of multiple paragraphs, where each paragraph is a text report recorded at different time points.

You are also provided with:
-A behavior family label.
-A brief description of this family.

Explain briefly and directly why the log belongs to the behavioral family, focusing only on concrete evidence from the log that matches the description. Write the explanation in a concise and professional style.

### Input
LOG: [*log*]
Behavioral Family: [*family name*]
Description of [*family name*]: [*family description*]

---

**Prompt for Unknown Attack Family**

---

You are a professional cybersecurity analyst. You are given a log consisting of multiple paragraphs, each recorded at different time points.

The log contains suspicious activity, but the type of attack is unknown. Analyze the log and provide a concise, professional explanation of why this log indicates an attack, focusing only on concrete evidence from the log itself.

### Input
LOG: [*log*]

---

structured yet heterogeneous, where each event contains multiple fields describing system state at a given moment, and sequences of events reflect the evolution of system behavior.

To address the limitations of existing public EDR datasets in terms of scale, diversity, and behavioral detail, we developed a data collection tool for real-time monitoring and capture of system activities. Specifically, we constructed multiple realistic enterprise environments, and automated scripts executed well-defined attack scenarios, including malware injection, privilege escalation, and lateral movement, enabling the capture of comprehensive execution traces of real-world malicious behaviors. Raw data was collected via the Windows kernel callback mechanism and stored in JSON format in real time, ensuring fine-grained and complete behavioral records. To support high-throughput data acquisition and real-time processing, a non-paged memory pool and circular buffering were employed to optimize event caching and transmission.

All events and their corresponding behavior chains were meticulously labeled by security experts to ensure annotation accuracy. The resulting dataset contains over 8 million event records, covering normal activities as well as 172 distinct real-world malicious families. For our experiments, the 20 most frequent behavior families (including normal activities) are used to construct standard training and testing sets, while the remaining 153 families serve as unknown attack scenarios.

Compared to existing public EDR datasets, EDR8M-20R combines large scale, event completeness, fine-grained behavioral chains, and high-quality expert annotations, making it one of the most complete datasets available for studying EDR detection and reasoning tasks.

## E. Dataset Release and Access Statement

To address potential dual-use concerns and ensure dataset privacy, the release of EDR8M-20R will follow a gated access policy. Researchers must apply for access and agree to terms that restrict use to legitimate research purposes, prohibit any attempts to reverse-engineer detection strategies, and require adherence to responsible disclosure practices. This approach helps prevent the dataset, along with the prompt designs and model code, from being misused to identify which behavioral signals are relied upon for detection.

All logs have been carefully anonymized to remove sensitive identifiable information, including usernames, hostnames, directory paths, and other system-specific identifiers. Encryption techniques are used to replace sensitive fields while keeping the overall semantic structure and statistical patterns intact, so that models can still learn meaningful behavior sequences without exposing private information.

In addition, the data collection and release procedures have been reviewed and approved by our institutional ethics committee. By combining controlled access, rigorous anonymization, and legal/ethical oversight, the dataset preserves analytical value for research while minimizing risks to privacy and potential misuse.

## F. Implementation Details of the Baseline Methods

In this work, we compare several state-of-the-art machine learning and deep learning methods in computer security, as well as recent large language models (LLMs). The implementation details of each method are described below.

**LGTF** (Kumar et al., 2022) is a malware detection method that leverages texture features extracted from images for effective pattern classification across multiple benchmark datasets. To adapt LGTF to our EDR dataset, we uniformly sampled 128 events per sample and transformed the logs into image sequences. Each character in an event is encoded in UTF-8 and mapped to a pixel in a predefined $56 \times 56$ image, with pixels arranged according to a fixed key order. Following the optimal configuration reported in the original work, we extract global features from each image, concatenate them, and apply the Bag-of-Visual-Words (BoVW) algorithm for dimensionality reduction. Finally, a Support Vector Machine (SVM) classifier is trained with tuned hyperparameters to achieve optimal performance.

**ADE** is a generalizable malicious URL detection method proposed by (Tsai et al., 2024). In our experiments, we adopted the MalConv baseline model (Catanzaro & Nicholas, 2018) following the implementation details provided in (Tsai et al., 2024), and applied the Adversarial Debiasing Embedding (ADE) training strategy for performance evaluation. All other hyperparameters remain consistent with the original configuration.

**DrSec** (Sharif et al., 2024) is a malicious behavior detection method based on pre-trained language models. Following the original implementation, we adopted the RoBERTa architecture to extract features from log text. To address the input context length limitation of RoBERTa, a sliding window strategy is applied for sequence segmentation. A shallow neural network classifier is attached to the output layer for end-to-end fine-tuning. All hyperparameters are kept consistent with the original settings, and we report the best evaluation results.

**DeepSeek-R1** (Guo et al., 2025) is a general-purpose reasoning model that achieves strong performance across various domains, surpassing OpenAI o1 (Jaech et al., 2024) on several benchmark tasks. Due to its substantial computational demands, we evaluated the full 671B-parameter model in a zero-shot setting. The input context length is set to the model's maximum of 128K tokens, with a temperature of 0.9 and a top-$p$ of 0.5. To constrain predictions to a predefined set of behavioral categories, all candidate family names are appended to the input prompt.

**Qwen3-235B-A22B** (Yang et al., 2025) is a large language model developed by the Qwen team with mixture-of-experts architectures. We evaluate it in a zero-shot setting using the same decoding configurations and prompts as DeepSeek-R1.

**Gemini-2.5-Pro** (Comanici et al., 2025) is a multimodal large language model developed by Google DeepMind. It demonstrates strong capabilities in reasoning, multimodal understanding, and long-context processing. We evaluate it in a zero-shot setting using the same hyperparameter configuration and prompts as DeepSeek-R1.

**GPT-5** (Singh et al., 2025) is a proprietary large language model developed by OpenAI. It exhibits strong general-purpose reasoning and instruction-following abilities across a wide range of tasks. We evaluate it in a zero-shot setting under the same evaluation protocol as DeepSeek-R1.

**Qwen2.5-1.5B/7B-Instruct** (Yang et al., 2024) is an open-source general-purpose language model. Through supervised fine-tuning, it has been successfully applied to various domain-specific tasks. In our experiments, we fine-tune the model using the same training set and perform inference with a 128K token context length. All other hyperparameters are kept consistent with those described in the experimental setup to ensure comparability and reproducibility.

**LLaMA3.1-8B-Instruct** (Grattafiori et al., 2024) is an open-source general-purpose language model developed by Meta AI, showing strong performance and generalization in various NLP tasks. We fine-tuned LLaMA3.1-8B-Instruct on the same training data, using parameter configurations identical to those used for Qwen2.5-1.5B/7B-Instruct to ensure fair comparison and reproducibility. We also perform inference at a 128K token context length.

*Table 11.* Prompts used for rationale evaluation in terms of Coherence, Consistency, and Completeness.

| Metric | Prompt Description |
|---|---|
| Coherence | You are an expert evaluator for large language model reasoning. Your task is to assess the **coherence** of the given rationale.
Coherence means that the rationale should be internally logical, step-by-step, and free from contradictions or abrupt jumps.

### Evaluation Criteria
1. Highly coherent: The rationale is well-structured and flows logically.
2. Moderately coherent: The rationale is mostly logical but has minor gaps, redundancies, or unclear transitions.
3. Poor coherence: The rationale is disorganized, contradictory, or difficult to follow.

### Input
- Model's Rationale: [*rationale*]

### Output Format
Provide:
- A short explanation of your judgment (2-3 sentences).
- A rating on a 1-5 scale (1 = very poor coherence, 5 = excellent coherence). |
| Consistency | You are an expert evaluator for large language model reasoning. Your task is to assess the **consistency** between the given rationale and the predicted answer.
Consistency means that the rationale should logically support the answer, without contradictions.

### Evaluation Criteria
1. Fully consistent: The rationale clearly and logically leads to the answer.
2. Partially consistent: The rationale provides partial support for the answer, but has gaps or irrelevant reasoning.
3. Inconsistent: The rationale contradicts the answer or fails to support it logically.

### Input
- Model's Answer: [*answer*]
- Model's Rationale: [*rationale*]

### Output Format
Provide:
- A brief explanation of your judgment (2-3 sentences).
- A rating on a 1-5 scale (1 = fully inconsistent, 5 = fully consistent). |
| Completeness | You are an expert evaluator for large language model reasoning. Your task is to assess the **completeness and alignment** of the model-generated rationale compared to a reference rationale.
Completeness means that the model's rationale should include the key reasoning steps from the reference rationale, without omitting essential parts.

### Evaluation Criteria
1. Highly aligned: The model rationale includes all key reasoning steps and closely matches the reference rationale.
2. Partially aligned: The model rationale includes some but not all key reasoning steps.
3. Misaligned: The model rationale misses most key reasoning steps or introduces unrelated/incorrect reasoning.

### Input
- Reference Rationale: [*reference_rationale*]
- Model's Rationale: [*predict_rationale*]

### Output Format
Provide:
- A short explanation of your judgment (2-3 sentences).
- A rating on a 1-5 scale (1 = very poor alignment, 5 = highly aligned). |

## G. Prompt for Rationale Evaluation

In our study, we evaluate the quality of generated rationales along three dimensions: **Coherence**, **Consistency**, and **Completeness**. This evaluation is particularly critical in the context of malicious behavior detection, where end users (e.g., security analysts) must not only trust the final predictions but also rely on interpretable reasoning processes to validate or further investigate the system's decisions.

Specifically, *Coherence* measures the logical flow and internal readability of a rationale, *Consistency* assesses whether the rationale faithfully supports the predicted label, and *Completeness* captures the degree to which the rationale aligns with a reference rationale.

To ensure reproducibility and reduce subjective bias, we employ multiple advanced language models as automatic evaluators, including GPT-5 (Singh et al., 2025), Gemini2.5-Pro (Comanici et al., 2025), and Qwen3-235B-A22B (Yang et al., 2025). Table 11 presents the prompts used for evaluation. Each prompt instructs the evaluators to provide both a concise textual justification (2–3 sentences) and a numerical rating on a 1–5 Likert scale. Using multiple evaluators helps mitigate model-specific biases and supports a more robust assessment of reasoning quality.

## H. Statistical Reliability of Results

To evaluate the statistical robustness of WatchLog, we conducted five independent runs using different random seeds and report the results as mean $\pm$ standard deviation in Table 12. The results exhibit consistently low variance across both detection and reasoning metrics. In particular, the standard deviations of Overall Accuracy (OA) and Rationale Completeness (Comp.) are only 0.78 and 1.1, respectively, indicating stable and reproducible performance under different random initializations.

*Table 12.* Performance of WatchLog over five independent runs (mean $\pm$ std).

| Method | BA↑ | R↑ | FA↓ | OA↑ | Cohe.↑ | Cons.↑ | Comp.↑ |
|---|---|---|---|---|---|---|---|
| WatchLog | 99.7$\pm$0.2 | 99.8$\pm$0.2 | 3.0$\pm$2.4 | 89.4$\pm$0.78 | 90.9$\pm$1.4 | 92.5$\pm$1.5 | 66.6$\pm$1.1 |

## I. Analysis of the Mathematical Properties and Computational Complexity of Each Stage

### I.1. Stage 1: Event-to-Image via kv2Embedding Transformation

Given an event $E$ decomposed into key–value pairs $\{\mathbf{x}_m = (k_m, v_m)\}_{m=1}^M$ and token embeddings $e_{m\ell} \in \mathbb{R}^d$, the kvEmbedding layer produces

$$\tilde{e}_m = \sum_{\ell=1}^{L_m} \alpha_{m\ell}\, e_{m\ell}, \qquad \alpha_{m\ell} = \frac{\exp(W^\top e_{m\ell})}{\sum_{u=1}^{L_m} \exp(W^\top e_{mu})},$$

with $W \in \mathbb{R}^d$. Two elementary but important properties follow immediately from this definition. First, $\tilde{e}_m$ is a convex combination of the tokens $\{e_{m\ell}\}$; consequently $\tilde{e}_m$ lies in the convex hull of the token vectors and in particular its norm is bounded by the largest token norm, i.e. $\|\tilde{e}_m\| \leq \max_\ell \|e_{m\ell}\|$. Second, the mapping from the multiset $\{e_{m\ell}\}_{\ell=1}^{L_m}$ to $\tilde{e}_m$ is permutation-invariant with respect to the token ordering: $\tilde{e}_m$ depends only on the multiset of token vectors and not on their sequence index, because both the logits $W^\top e_{m\ell}$ and the convex combination are pointwise functions of each token. These properties imply that kvEmbedding provides a stable, order-agnostic pooling of variable-length values into a fixed-dimensional representation; stability here is a direct consequence of the softmax normalization (bounded coefficients) and differentiability.

From a computational viewpoint, computing $\{\tilde{e}_m\}_{m=1}^M$ requires a linear pass over the tokens: each token needs the dot product $W^\top e_{m\ell}$ and a weighted sum, so the per-event cost scales as $O\big(\sum_{m=1}^M L_m \cdot d\big)$. After projection into the ViT space and reshaping the $M$ vectors into $\tilde{E} \in \mathbb{R}^{c \times h \times w}$ (with $M = hw$), the visual footprint of a single event is $O(c \cdot h \cdot w)$ in activation memory.

The image–description objective is an autoregressive negative log-likelihood

$$\mathcal{J}_{IEA} \;=\; -\sum_{l=1}^{L_I} \log P_{\theta'}(\mathcal{D}_l \mid \mathcal{D}_{<l}, E).$$

Viewed probabilistically, minimizing the expected value of $\mathcal{J}_{IEA}$ over the data distribution is equivalent to minimizing the conditional Kullback–Leibler divergence between the empirical conditional distribution of descriptions and the model conditional distribution induced by $\tilde{E}$. Formally,

$$\mathbb{E}_{(E,\mathcal{D})\sim\mathrm{D}}\big[\mathcal{J}_{IEA}\big] = \mathbb{E}_E\big[\, H\big(p_{\mathrm{data}}(\cdot \mid E)\big)\big] + \mathbb{E}_E\big[\, \mathrm{D}_{\mathrm{KL}}\big(p_{\mathrm{data}}(\cdot \mid E)\,\big\|\,p_{\theta'}(\cdot \mid \tilde{E})\big)\big],$$

where $p_{\mathrm{data}}(\mathcal{D} \mid E)$ denotes the (possibly stochastic) distribution of natural-language descriptions for event $E$. Thus training with $\mathcal{J}_{IEA}$ explicitly encourages the learned visual encoding $\tilde{E}$ to be predictive of human-interpretable descriptions in the sense of minimizing the conditional KL divergence. Practically, this provides a semantically rich inductive bias: the encoder must produce features that preserve the aspects of the event that are informative for language generation. Because the kvEmbedding is permutation-invariant and norm-bounded, it yields numerically stable gradients and avoids exploding activations in this alignment step.

This stage therefore constructs compact, semantically grounded event images $\tilde{E}$ with controlled computational cost and a clear maximum-likelihood interpretation; these images serve as the atomic visual primitives that the temporal stage will aggregate.

## I.2. Stage 2: Log-Seq → Video via temporal cross-attention

Stacking the per-event images produces a video-like tensor $\mathcal{E} = [\tilde{E}_1, \ldots, \tilde{E}_T] \in \mathbb{R}^{T \times c \times h \times w}$. To reduce temporal redundancy we form a small set of queries $Q = \mathcal{E}_{\mathrm{pool}} \in \mathbb{R}^{t \times c \times h \times w}$ (with $t \ll T$) and apply cross-attention at every spatial location:

$$\tilde{\mathcal{E}}_{i,j} \;=\; \mathrm{Softmax}\!\left(\frac{Q_{i,j}W_Q (K_{i,j}W_K)^\top}{\sqrt{c}}\right)(V_{i,j}W_V), \qquad K, V = \mathcal{E},$$

so that for each pixel coordinate $(i, j)$ the condensed vectors $\tilde{\mathcal{E}}_{i,j}$ are linear combinations of the original temporal vectors $\{V_{i,j}(\tau)\}_{\tau=1}^{T}$. Consequently, at every spatial location the condensed representation lies in the linear span of the $T$ original temporal vectors; in matrix terms the cross-attention realizes a rank-at-most-$t$ projection of the $T$-frame sequence at each pixel. The rank bound implies a monotone relationship between the transformed size $t$ and representational capacity: increasing $t$ enlarges the span that can be represented and thereby reduces the approximation error (measured, e.g., in squared reconstruction error) of temporal dynamics that can be captured at that pixel. Conversely, choosing $t \ll T$ provides a controlled low-rank approximation that removes redundancy and focuses capacity on salient temporal patterns.

The computational cost of forming $\tilde{\mathcal{E}}$ dominates along three dimensions: the number of frames $T$, the condensed frame count $t$, and the spatial and channel dimensions $h, w, c$. A direct implementation computing attention scores for every $(i, j)$ incurs cost proportional to $O(h \cdot w \cdot c \cdot t \cdot T)$ for the main score multiplications, and requires activation storage roughly $O(T \cdot c \cdot h \cdot w)$ for keys and values. Using $t \ll T$ therefore translates into substantial arithmetic and memory savings in downstream processing, while the pixelwise attention ensures that spatial consistency is preserved (temporal aggregation is local in the spatial index and thus does not mix unrelated pixel locations).

The video–log objective is again an autoregressive NLL

$$\mathcal{J}_{VLA} \;=\; -\sum_{l=1}^{L_V} \log P_{\theta''}(\mathcal{S}_l \mid \mathcal{S}_{<l}, \mathcal{L}),$$

and the same KL-decomposition as in Stage 1 holds: minimizing $\mathcal{J}_{VLA}$ pushes the model conditional $p_{\theta''}(\cdot \mid \tilde{\mathcal{E}})$ toward the empirical conditional distribution of log narratives. Because $\tilde{\mathcal{E}}$ is formed from event-level encodings that were themselves grounded in language in Stage 1, the cross-attention step has the dual effect of (i) transforming temporal information into a compact video representation and (ii) preserving the semantic axes that are relevant for language generation. In particular, temporal cross-attention is time-aware (the key positions carry time indices) so the model can learn order-sensitive dynamics rather than a mere bag-of-frames summary.

Practically, this stage thus produces a temporally condensed but semantically aligned video tensor $\tilde{\mathcal{E}}$ that balances representational fidelity (controlled by $t$) against computational footprint, and it prepares a temporally coherent input for the final supervised stage.

### I.3. Stage 3: Supervised fine-tuning

In fine-tuning we train the multimodal model end-to-end to produce a family label $y$ and a natural-language rationale $r$. The joint autoregressive objective is

$$\mathcal{J}_{SFT} = -\sum_{l=1}^{L} \log P_{\theta^*}\big((y,r)_l \mid (y,r)_{<l}, (\mathcal{L}, q)\big).$$

Using the chain rule for probabilities the joint target factorizes as $P(y, r \mid \tilde{\mathcal{E}}, q) = P(y \mid \tilde{\mathcal{E}}, q) \cdot P(r \mid y, \tilde{\mathcal{E}}, q)$. Minimizing $\mathcal{J}_{SFT}$ therefore simultaneously improves the discriminative component $P(y \mid \cdot)$ and the conditional explanation model $P(r \mid y, \cdot)$. From a learning-theory perspective this is a form of multi-task or auxiliary-task training in which rationales act as a task-consistent regularizer: the need to generate coherent, label-consistent explanations constrains the function class and can reduce overfitting on the classification objective alone. In probabilistic terms, the expected supervised loss decomposes into the target entropy plus a KL term that measures the discrepancy between the empirical conditional distribution of $(y, r)$ and the model distribution; minimizing it yields a maximum-likelihood estimate for the joint target under the model family.

Gradient signals from $\mathcal{J}_{SFT}$ flow into both the language model parameters and the visual encoder that produced $\tilde{\mathcal{E}}$, thereby refining visual features to be discriminative for the final task while maintaining semantic alignment learned during pretraining. Because Stage 1 and Stage 2 already shaped $\tilde{\mathcal{E}}$ to encode human-interpretable structure, the supervised optimization typically requires fewer task-specific examples to reach a given error level compared to training from random initialization: the pretraining stages provide an informative initialization (a form of data-dependent prior) that reduces effective sample complexity in standard transfer-learning regimes.

On convergence and practical behavior: each objective $\mathcal{J}_{IEA}, \mathcal{J}_{VLA}, \mathcal{J}_{SFT}$ is a sum of negative log-probabilities and thus is amenable to stochastic gradient optimization; while nonconvexity precludes global optimality guarantees, under common smoothness and bounded-variance assumptions gradient methods converge to stationary points. In practice, a stagewise schedule (first Stage 1, then Stage 2, finally Stage 3 fine-tuning) is well matched to the probabilistic decomposition above because it first enforces local semantic alignment (events), then temporal coherence (videos), and finally task discrimination and explanation.

## J. Supplementary Numerical Results

To complement the visualizations in the main text, we provide the corresponding numerical values in tabular form. These tables ensure reproducibility and enable precise inspection of our results. Each table corresponds to one figure in the main body, and provides the exact scores underlying the plots.

### J.1. Ablation on Temporal Value $t$ (Figure 3, top row)

Table 13 provides the raw numbers for the ablation study on temporal transformation length $t$, corresponding to the top row of Figure 3. We vary $t \in \{32, 48, 64, 80\}$ and report both detection metrics (False Alarm, Overall Accuracy) and reasoning scores (all rescaled to 0–100). The visualization in the main text illustrates overall trends, whereas this table allows precise comparison across different $t$ settings.

*Table 13.* Ablation study on the effect of temporal value $t$ (corresponding to Figure 3, top row). Detection metrics are in percentage (%), and reasoning metrics are rescaled to 0–100.

| $t$ **Value** | **FA** $\downarrow$ | **OA** $\uparrow$ | **Cohe.** $\uparrow$ | **Cons.** $\uparrow$ | **Comp.** $\uparrow$ |
|---|---|---|---|---|---|
| 32 | 10 | 87.5 | 94.0 | 95.2 | 66.7 |
| 48 | 5 | 89.5 | 92.8 | 94.1 | 66.6 |
| 64 | 5 | 90.3 | 93.2 | 95.0 | 68.1 |
| 80 | 5 | 89.8 | 93.1 | 94.2 | 66.7 |

## J.2. Ablation on Scaling Factor (Figure 3, bottom row)

Table 14 corresponds to the bottom row of Figure 3, which studies the effect of scaling event-level cross-attention modules by different factors. We report detection and reasoning metrics for the default (×1) and scaled variants (×2, ×3, ×4). The table complements the main-figure visualization with exact values, making clear the slight performance trade-offs introduced by scaling.

*Table 14.* Ablation study on the effect of scaling the event-level cross-attention modules (corresponding to Figure 3, bottom row). Detection metrics are in percentage (%), and reasoning metrics are rescaled to 0–100.

| Scaling Factor | FA↓ | OA↑ | Cohe.↑ | Cons.↑ | Comp.↑ |
|---|---|---|---|---|---|
| 1 | 5 | 90.3 | 93.2 | 95.0 | 68.1 |
| ×2 | 5 | 88.5 | 92.9 | 94.6 | 65.8 |
| ×3 | 5 | 88.5 | 94.1 | 95.8 | 65.8 |
| ×4 | 5 | 87.3 | 92.2 | 94.5 | 64.9 |

# K. Spatial Sensitivity Analysis and Visualization for Log Predictions

## K.1. Spatial Arrangement Sensitivity

To investigate the robustness of our method to spatial arrangement sensitivity, we perform an additional study with two experimental settings involving randomly permuted patch layouts. In the first setting, the model is trained with randomly permuted patches but tested using the default layout (alphabetically by key). In the second setting, both training and testing involve randomly permuted patches, and we report the averaged results over five runs.

As shown in Table 15, the results demonstrate that patch permutation substantially reduces the false alarm rate, albeit with a slight decrease in overall accuracy. For the reasoning-quality metrics, applying random permutation only during training improves *Coherence* and *Consistency*, with a modest decrease in *Completeness*. When permutation is applied during both training and testing, the reductions in reasoning-quality metrics remain within acceptable bounds.

Overall, these results demonstrate that WatchLog is robust to spatial arrangement variations, maintaining high performance even when patch layouts are randomized. The minor fluctuations observed further highlight the flexibility and resilience of our approach, making it effective even when the spatial arrangement of input patches changes.

*Table 15.* Ablation study on spatial arrangement sensitivity.

| Method | FA ↓ | OA ↑ | Cohe. ↑ | Cons. ↑ | Comp. ↑ |
|---|---|---|---|---|---|
| WatchLog (default) | 5.0 | 90.3 | 93.6 | 95.0 | 68.5 |
| WatchLog (perm. train only) | 0.0 | 88.5 | 94.2 | 96.2 | 67.1 |
| WatchLog (perm. train & test) | 1.0 | 88.0 | 92.9 | 94.6 | 64.8 |

## K.2. Visualization of Log Analysis

To provide an intuitive illustration of how our framework interprets malicious behaviors, we present a visualization of log analysis in Table 16 to Table 18. For clarity and readability, certain field values (e.g., MD5 values and file paths) are partially masked with ". . ." placeholders. This visualization shows that our approach not only achieves accurate detection, but also delivers interpretable evidence of how malicious behaviors evolve over time.

*Table 16.* Visualization of our model's reasoning on a log sample from the *Win32_Worm.FakeFolder* attack. Text highlights with the same color indicate identical rationale semantics.

---

**Input Log: [*An ultra-long log sequence...*]**

---

Ground-Truth family: Win32_Worm.FakeFolder

Ground-Truth rationale:

-Self-replication/propagation: On 2024-04-17, a PE file named 9d82…e9e2.exe is written to D:\sample\9d82….e9e2.exe, indicating copying a main payload to another location for propagation.

-Persistence via Run keys: Recurrent AsepValueUpdate events show the registry Run key HKEY_LOCAL_MACHINE\...\Windows\CurrentVersion\Run updated to load system.exe (e.g., C:\Windows\system32\system.exe or C:\Windows\syswow64\system.exe) across multiple dates (May–June 2024). This demonstrates startup persistence typical of FakeFolder behavior.

-DLL-based execution chain: Multiple LoadImage and AsepValueUpdate entries show rundll32.exe executing C:\Windows\system32\rlfmb.dll Exucute and loading other DLLs (e.g., ifaja.dll, rlfmb.dll). This indicates payload execution via DLLs loaded through Rundll32, a common technique for FakeFolder variants.

-Deceptive/masquerading and complex process chain: The event sequences consistently show long, multi-step process chains (smss.exe → winlogon.exe → userinit.exe → explorer.exe, etc.) preceding the malicious actions, consistent with a worm attempting to blend into system startup and user sessions.

In sum, the logs show self-replication attempts, repeated Run-key persistence, and DLL-based execution paths, all of which map to the described behavior of Win32_Worm.FakeFolder.

---

Our predictions: Win32_Worm.FakeFolder

Our rationale:

-Self-replication/propagation indicators: The same binary (7f…9f.exe) is repeatedly created and executed from D\:\sample\with different timestamps, suggesting replication to local drives rather than just memory. The same MD5 hash is used across events, confirming a single binary propagating across the host.

-Masquerading and persistence via startup mechanisms: Multiple AsepValueUpdate events show the registry Run key (HKEY_LOCAL_MACHINE\...\Microsoft\Windows\CurrentVersion\Run) being set to launch C:\Windows\system32\system.exe, i.e., a persistence mechanism designed to auto-start on logon. This is a classic FakeFolder tactic to ensure re-execution after reboot.

-Use of rundll32 to execute payloads: Repeated LoadImage events show rundll32.exe loading and executing wgbhf.dll (Exucute), with command lines like C:\Windows\system32\rundll32.exe Exucute (and variants). This DLL loading pattern is typical for payload deployment executed via legitimate system processes.

-Process chain and stealth/persistence pattern: The events describe a long process chain (smss.exe → winlogon.exe → userinit.exe → explorer.exe → system.exe → rundll32.exe, etc.) with high integrity levels and extensive cleanup-write actions, consistent with attempts to establish persistence and concealment.

In summary, the log shows self-replication to local drives, Run-key persistence for auto-start, and rundll32-based payload execution, all characteristic of Win32_Worm.FakeFolder.

*Table 17.* Visualization of our model's reasoning on a log sample from the *Win32_TrojanDownloader.Upatre* attack. Text highlights with the same color indicate identical rationale semantics.

---

**Input Log: [*An ultra-long log sequence...* ]**

---

Ground-Truth family:   Win32_TrojanDownloader.Upatre

Ground-Truth rationale:

  -Masquerading dropper: Numerous entries depict executables named WinWord.exe (Office 2007) and related temporary variants (e.g., 86EC1D..., DE77.tmp, DC92.tmp, 4F71.tmp, etc.) being created and executed from the user/admin Temp paths. This pattern fits Upatre's practice of dropping a loader under a trusted-looking name to bypass user suspicion.

  - On-disk payload writes in Temp: Repeated PE write events (PeFileWritten) with FILE_OP_FLAG_WRITTEN_ON_CLEANUP in C:\Users\admin\AppData\Local\Temp and similar Temp folders indicate multiple payloads being written during cleanup phases, consistent with a downloader staging subsequent stages.

  - Multi-stage process chains: The log shows long chains of temporaries and chained processes (e.g., A31E.tmp → A69A.tmp → 3BD4.tmp → 2D4D.tmp, etc.), a hallmark of downloader activity that drops and executes successive payloads.

  - Elevated/hidden execution context: Many processes run with high integrity and network level 70, and several are launched without command lines, suggesting stealthy stages and potential C2/download activity.

In short, the observations match Upatre's typical behavior pattern: initial access via a disguised dropper, on-disk multi-stage payload delivery in Temp, extensive use of chained temporary executables, and implied network-based payload retrieval, all within a Word/Office context.

---

Our predictions:   Win32_TrojanDownloader.Upatre

Our rationale:

  - On-disk dropper activity in Temp with obfuscated filenames and repeated PE writes (PeFileWritten) in a long process chain.

  - The log shows numerous temporary PE files being created and written in the user Temp folder (for example: 9A1.tmp, 3D21.tmp, 62B3.tmp, 7A1.tmp, etc.), each associated with WinWord.exe from the 2007 Office suite. These files are written to C:\Users\admin\AppData\Local\Temp\ and then later written to disk, consistent with Upatre's behavior of dropping multiple staged PE payloads on disk.

  - Multi-stage, long process chains centered on WinWord.exe from Office 2007.

  - A long chain of temporaries (e.g., 9A1.tmp → 3D21.tmp → 62B3.tmp → 7A1.tmp → 1E08.tmp → 22B.tmp → 3034.tmp → 337.tmp → 3691.tmp → 39E.tmp → 42B.tmp → 449.tmp, etc.) all referencing WinWord.exe from the 2007 Office suite. This pattern matches Upatre's multi-stage downloader that fetches and executes a second-stage payload by abusing a chain of temporary processes.

  - High integrity, network-capable processes forming the downloader chain.

  - The involved processes in the chain (WinWord.exe) appear with high integrity and a network level of 70 across many events, consistent with Upatre's behavior of network-enabled download/drop operations and subsequent payload execution.

  - Cleanup-time PE writes (FILE_OP_FLAG_WRITTEN_ON_CLEANUP) indicative of staged payloads.

  - Numerous PeFileWritten events show PE files being written during cleanup (for example: 9A1.tmp writes 9D07.tmp; 3D21.tmp writes 3E11.tmp; and so on).  This aligns with Upatre's technique of dropping and finalizing second-stage payloads as cleanup occurs.

In short, the log exhibits Upatre-like behavior: extensive on-disk, multi-stage PE drops in Temp, a long chain of Office-temporary processes referencing WinWord.exe, and cleanup-time file writes that together reflect the downloader-and-payload-fetch pattern of Win32_TrojanDownloader.Upatre.

---

*Table 18.* Visualization of our model's reasoning on a log sample from the *Worm.Win32.Scar* attack. Text highlights with the same color indicate identical rationale semantics.

---

**Input Log: [An ultra-long log sequence. . . ]**

---

Ground-Truth family:  Worm.Win32.Scar

Ground-Truth rationale:

-Self-propagation/replication signals in a single host:

- Repeated creation and execution of numerous Sysqem*.exe binaries located in C:\Users\admin\AppData \Local\Temp\ across many timestamps, each with high integrity and network level 70. This pattern is consistent with Scar's self-replicating payloads and stagedDroppers that proliferate multiple components on a host.

-File-based propagation with cleanup writes (typical Scar tactic):

- Frequent PE file writes in Temp directories labeled as PeFileWritten with FILE_OP_FLAG_WRITTEN_ON_CLEANUP, often accompanied by long chains of Sysqemxxx processes.

This indicates dropping and cleaning up multiple executable components rather than relying on registry changes or explicit network actions.

- Logs show extended chains from system processes (smss.exe, winlogon.exe, explorer.exe) through multiple Sysqem-related executables, suggesting a staged, multi-component infection flow typical of Scar's approach to persistence and payload deployment.

-Lack of explicit user-driven commands or targeted network actions:

While network level values exist, the events emphasize file drops and process chaining over command-line payloads or registry edits, aligning with Scar's basic self-propagation behavior rather than a targeted, user-initiated intrusion.

In sum, the behavior—large-scale, internal propagation via many Sysqem*.exe Dropper/Payload components in Temp with cleanup-file writes, and long process chains originating from core Windows processes—matches the Scar worm's core traits (self-propagation, self-replication, staged drops, lightweight persistence) as described for Worm.Win32.Scar.

---

Our predictions:  Worm.Win32.Scar

Our rationale:

- The log shows a long, multi-stage dropper chain with numerous Sysqem*.exe payloads dropped into the Temp folder (C:\Users\ admin\AppData\Local\Temp) and then further staged executables (e.g., Sysqemrjedk.exe, Sysqemtz. . . etc.). This pattern is characteristic of Scar's self-propagating worm behavior, where a small dropper copies itself into accessible locations and spawns additional payloads for propagation.

- Repeated file writes of PE files to the Temp directory during cleanup (FILE_OP_FLAG_WRITTEN_ON_) CLEANUP indicate the drop-and-persist/evade approach typical of Scar-like worms, which survive to later stages by writing new executables to disk.

- The process chains trace through system processes (smss.exe → winlogon.exe → explorer.exe → multiple Sysqem*.exe processes) and high-integrity, high-network-level activity across the chain. This aligns with Scar's propagation and persistence through Windows startup/Explorer lineage.

- Although explicit USB/removable-media activity is not shown, the pervasive local drop-and-run behavior (new Sysqem*.exe created in Temp, then further child processes) matches Scar's described pattern of self-propagation and persistence on connected hosts.

In short, the observed self-replicating dropper chain in Temp with multi-stage process chains and cleanup-file writes is consistent with the behavioral profile of Worm.Win32.Scar.

---

