# OpenReview forum: "WatchLog: Efficient and Interpretable Event Reasoning for Endpoint Detection and Response Logs with Multimodal LLMs"
_ICML.cc/2026/Conference — ICML 2026 regular_

### Official Review · Reviewer_jY69 · 2026-03-11

**Soundness:** 3
**Presentation:** 2
**Significance:** 2
**Originality:** 2
**Overall Recommendation:** 3
**Confidence:** 2

**Summary:**

This paper introduces WatchLog, a novel multimodal framework for Endpoint Detection and Response (EDR). It converts ultra-long EDR logs into video-structured data through a three-stage process (event-to-image conversion, video-language modeling, and reasoning-aware fine-tuning). This approach enables large language models (LLMs) to efficiently process massive contexts, detect malicious behavior accurately, and provide interpretable reasoning for security analysts.

**Compliance With Llm Reviewing Policy:**

Affirmed.

**Final Justification:**

The lack of qualitative visualizations makes me uncertain about the model’s capability, so I maintain my rating.

**Key Questions For Authors:**

see weakness

**Limitations:**

yes

**Strengths And Weaknesses:**

**Strengths:**

Both the motivation and the method are relatively novel. By representing raw logs as structured video-like data, the paper enables scalable and expressive video-language modeling for endpoint behavior. The incorporation of LLMs also makes the model outputs more interpretable.

The paper provides a new dataset, which is a valuable contribution to the development of the EDR community.

---

**Weaknesses:**

**Limited contribution:**
The proposed benchmark appears to involve overly simple tasks. Since WatchLog and LLMs have already achieved performance above 90%, it is difficult for this benchmark to provide meaningful guidance for future research in this area.

**Writing issues:**
The paper lacks an overall framework diagram and does not clearly explain the differences between WatchLog and previous methods.

It also lacks visualizations of the generated images and generated videos.

**Insufficient experiments:**
Among the zero-shot models, only DeepSeek and Qwen are evaluated. The paper should also compare against stronger models, such as Gemini, GPT, and Claude.

In addition, the evaluation on existing EDR benchmarks is still insufficient. The experiments only include a subset of ATLASv2, which is not enough to fully demonstrate the effectiveness of WatchLog.

---

> ### Author Rebuttal · Authors · 2026-03-31
>
> Thank you for the detailed and constructive feedback. We appreciate the reviewer’s insights and address the main concerns—**benchmark difficulty**, **clarity of presentation**, and **additional evaluations**—point by point below.
>
> **R1: Benchmark Contribution and Saturation**
>
> We respectfully clarify that the >90% performance in Table 2 **reflects in-domain classification on our real-world EDR attack-defense dataset, and does not imply that the benchmark is saturated or lacks research value**. Such performance levels are commonly observed in security-related benchmarks. For example, malware detection datasets such as EMBER [1], fraud detection benchmarks such as ScamNet [2], and prior work on APT detection (e.g., Aly et al. [3] on DARPA datasets) report similarly high accuracy, while still serving as important testbeds for robustness and generalization research. High in-domain accuracy therefore does not indicate that the problem is fully solved, but instead highlights the need for more robust models.
>
> Importantly, **our benchmark introduces several practical challenges, including out-of-distribution (OOD) generalization (Table 3), ultra-long log reasoning (Figure 3), and rationale generation (Table 2)**. These settings leave substantial room for improvement and provide a meaningful testbed for evaluating truly robust security models in real-world environments.
>
> **R2: Clarity of Presentation and Visualizations**
>
> **Regarding the lack of an overall framework**, we would like to clarify that an **overall framework diagram is provided in Figure 2**, where each module is explicitly illustrated along with its corresponding training stage. The Related Work section (Sec. 2.1-2.4) systematically discusses differences between WatchLog and existing approaches, including traditional methods, deep learning methods, LLM-based methods, and long-context LLMs. To further improve clarity, **we will move Figure 2 earlier in the manuscript to enhance readability and strengthen the discussion of differences** between our method and existing approaches in the Related Work section.
>
> **Regarding visualization of the transformed video**, the log-to-video representation is a high-dimensional tensor (T×C×H×W) (Figure 2), where individual “pixels” correspond to embedded fields rather than actual image content. Direct visualization is therefore not informative for human interpretation. We will consider providing a dynamic demo on the project’s open-source homepage to illustrate the log-to-video transformation process.
>
> **R3: Comparison with Stronger LLMs**
>
> We agree and have **extended our experiments to include stronger frontier LLMs** (Gemini-2.5-Pro, GPT-5, Claude-sonnet-4.5) via API. The results show that while these models continue to exhibit performance limitations when handling EDR logs (similar to DeepSeek and Qwen in Table 2), especially in discriminating malicious behaviors, WatchLog maintains superior performance. We will include these comparisons in the revision.
>
> | | BA↑ | R↑ | FA↓ | OA↑ |
> |:-|:-|:-|:-|:-|
> | Gemini-2.5-Pro | 76.3 | 77.6 | 50.0 | 9.0 |
> | GPT-5 | 63.8 | 62.4 | 10.0 | 13.5 |
> | Claude-sonnet-4.5 | 45.3 | 42.4 | 0.0 | 10.0 |
> | WatchLog (Ours) | 99.8 | 100.0 | 5.0 | 90.3 |
>
> **R4: Evaluation on More Public Datasets**
>
> We use a subset of ATLASv2 because it specifically corresponds to EDR log data, aligning with our problem setting. **Other parts of ATLASv2 deviate substantially from practical EDR scenarios (Line 371-375)**. More broadly, in the early stage of our study, we conducted extensive investigations and found that publicly available EDR datasets are very limited, **which motivated our construction of EDR8M-20R**.
>
> To provide a comprehensive evaluation, in addition to ATLASv2 (Table 4), **we conduct experiments on EDR8M-20R under multiple complementary settings**: (i) a standard test set aligned with the training distribution (Table 2), (ii) an independently collected test set (Table 3), and (iii) out-of-distribution scenarios with unseen attack patterns (Table 3). Across all settings, WatchLog consistently achieves strong performance, demonstrating its robustness and generalization capability.
>
>
>
> *[1] Joyce, R. J., Miller, G., Roth, P., Zak, R., Zaresky-Williams, E., Anderson, H., ... & Holt, J. Ember2024-a benchmark dataset for holistic evaluation of malware classifiers. In Proceedings of the 31st ACM SIGKDD Conference on Knowledge Discovery and Data Mining V. 2. pp. 5516-5526, 2025.*
>
> *[2] Bitaab, M., Karimi, A., Lyu, Z., Mosallanezhad, A., Oest, A., Wang, R., ... & Doupé, A. Scamnet: Toward explainable large language model-based fraudulent shopping website detection. In Proceedings of the AAAI Conference on Artificial Intelligence. pp. 27841-27848, 2025.*
>
> *[3] Aly, A., Mansour, E., & Youssef, A. OCR-APT: Reconstructing APT stories from audit logs using subgraph anomaly detection and LLMs. In Proceedings of the 2025 ACM SIGSAC Conference on Computer and Communications Security. pp. 261-275, 2025.*

---

> > ### Author Rebuttal · Reviewer_jY69 · 2026-04-03
> >
> > Thank you for the authors’ response. However, I still have some questions.
> >
> > If stronger methods such as LGTF, ADE, and DrSec were trained on EDR8M-20R, would their performance improve further, enabling a fairer comparison?
> >
> > To what extent does WatchLog’s performance gain come from the model architecture itself, and to what extent does it come from the data?

---

> > > ### Author Response · Authors · 2026-04-06
> > >
> > > Thank you very much for the follow-up questions and for taking the time to further engage with our work—we really appreciate it.
> > >
> > > Regarding your first question, we would like to clarify that **the methods we compare (including LGTF, ADE, DrSec, Qwen2.5-1.5/7B, LLaMA3.1-8B, and WatchLog) are trained on the same EDR8M-20R training set and evaluated on the same test set throughout our experiments, so that all methods are evaluated under a consistent setting. Additional details on the training setups are also included in Appendix F for completeness.**
> > >
> > > For your second question, since all methods **share the same training and test data, the observed performance gains are not driven by differences in data, but instead stem mainly from the proposed framework design**. This is reflected in both the **methodological contributions** and the **empirical results** (e.g., Tables 2–5), where WatchLog consistently outperforms prior approaches under the same conditions.
> > >
> > > We hope this helps clarify the setup and our findings, and we are happy to provide any additional details if needed. If you feel it appropriate, we would appreciate any update that reflects this understanding.
> > >
> > > Thank you again for your time and thoughtful questions.

---

### Official Review · Reviewer_4rDu · 2026-03-13

**Soundness:** 2
**Presentation:** 2
**Significance:** 3
**Originality:** 3
**Overall Recommendation:** 4
**Confidence:** 2

**Summary:**

The manuscript proposed WatchLog to  represent raw logs as video-structured data, enabling scalable and expressive video-language modeling of endpoint behaviors. It is interesting to encode each event as a key-value-guided image and organize the resulting images into a video sequence.

**Compliance With Llm Reviewing Policy:**

Affirmed.

**Final Justification:**

This paper proposes WatchLog, a novel multimodal language modeling framework for EDR log analysis that addresses the scalability and generalization limitations of existing LLM-based log analysis methods. My initial review raised a core concern: the authors should add comparisons with dedicated long-context LLMs and state-of-the-art multimodal security analysis models, alongside engineering-level metrics (parameter scale, training/inference costs, deployment resource requirements) to better highlight WatchLog’s core advantages. After reviewing the authors’ detailed rebuttal, I confirm all core concerns have been resolved. The authors provided conclusive quantitative comparisons with long-context LLMs (Qwen-1.5B augmented with LongRoPE/Sliding Window Attention), demonstrating that WatchLog achieves superior system efficiency: for 1M token processing, it requires ~15.47GB GPU memory and ~1s latency, compared to 26.87–98.46GB memory and 20.49–259.30s latency for long-context baselines, with clear deployment scalability advantages (1 GPU vs. 3 GPUs for Qwen-7B). They also clearly distinguished WatchLog from state-of-the-art multimodal security models (Logprompt, LogLM, Srdc), highlighting that WatchLog is the first framework to introduce multimodal language modeling principles into EDR log analysis via a video-like structured log representation, overcoming the scalability and generalization limitations of prior methods that rely on raw LLM prompting or handcrafted features. The supplementary engineering metrics and baseline comparisons fully validate WatchLog’s practical deployment value and technical novelty, making the paper suitable for acceptance at ICML. I appreciate the authors’ thorough and evidence-based response to my initial concerns.

**Key Questions For Authors:**

I suggest the authors add comparisons with dedicated long-context LLMs and state-of-the-art multimodal security analysis models, along with engineering-level metrics such as parameter scale, training/inference costs and deployment resource requirements, to better highlight the core advantages of WatchLog.

**Limitations:**

Yes

**Strengths And Weaknesses:**

Its experiments well validate the framework's core innovations: dedicated tests prove the log-to-video transformation effective, and targeted ablation verifies the temporal cross-attention module's unique value for long-sequence EDR analysis, confirming the originality of the spatio-temporal modeling paradigm. However, there is no comparison with dedicated long-context and multimodal security models, and the ablation of core parameters and sub-modules fails to explore the performance boundaries and their core contribution points.

---

> ### Author Rebuttal · Authors · 2026-03-31
>
> We sincerely thank the reviewer for this valuable suggestion. We provide additional clarifications and comparisons to better highlight the engineering advantages of WatchLog from three perspectives: system-level efficiency metrics, long-context LLMs, and multimodal security models.
>
> **Efficiency comparison with WatchLog**. As shown in Table 5, **WatchLog achieves significantly better system efficiency compared to strong LLM baselines in terms of GPU memory and latency**. For processing 1M tokens, WatchLog requires approximately 49GB GPU memory and ~1s latency, whereas Qwen-7B requires about 215GB memory and 486s latency. This translates to 1 GPU (A100 80GB) for WatchLog versus at least 3 GPUs for Qwen-7B, demonstrating clear advantages in deployment scalability and real-world feasibility. To further highlight the practical advantages of our method, we will extend Table 5 in the Appendix to include a quantitative comparison of training cost and GPU requirements for deployment.
>
> **Comparison with long-context LLMs**. Representative methods such as YaRN, LongRoPE, Longformer, and hybrid attention can significantly enhance the long-context capabilities of LLMs, but they still rely on tokenizing the full text sequence, leading to inherently high memory and compute costs at inference. To further validate this, **we conducted additional experiments** with Qwen-1.5B augmented with LongRoPE or Sliding Window Attention (SW=32K) in half of the layers.
>
> | ||| GPU-MU (GB) | | | | |TTFT (s) | |
> |:-|:-:|:-:|:-:|:-:|:-:|:-:|:-:|:-:|:-:|
> | | |256K | 512K | 1024K | | |256K | 512K | 1024K |
> | Qwen2.5-1.5B-LongRoPE | |26.87 | 50.77 | 98.46 | | |20.49 | 78.81 | 259.30 |
> | Qwen2.5-1.5B-SWA | | 16.56 | 28.60 | 52.68 | | |11.56 | 41.93 | 133.80 |
> | WatchLog (Ours) | | **15.47** | **26.81** | **48.61** | | |**0.69** | **0.79** | **1.11** |
>
> The results consistently show that long-context LLMs incur substantially higher resource consumption and latency than WatchLog, especially as input length increases. This suggests that our advantage stems from the modeling paradigm rather than solely relying on architectural scaling of token-based modeling.
>
> **Comparisons with multimodal security models**. The approaches most relevant to our study are large language models applied to system security and log analysis, including Logprompt [1], LogLM [2], and SRDC [3]. These approaches either rely on direct prompting of LLMs over raw logs, which suffers from poor scalability for long EDR sequences, or depend on handcrafted feature pipelines, limiting generalization and adaptability.
>
> In contrast, WatchLog is, to the best of our knowledge, **an initial framework that introduces multimodal language modeling principles into EDR log analysis by representing raw logs in a video-like structured format**. This design enables hierarchical spatio-temporal modeling of log behaviors, achieving both strong expressiveness and improved computational efficiency, making it more practical for real-world deployment.
>
> We thank the reviewer again for this helpful suggestion, and we hope our response addresses your concern.
>
>
> *[1] Liu, Y., Tao, S., Meng, W., Yao, F., Zhao, X., & Yang, H. Logprompt: Prompt engineering towards zero-shot and interpretable log analysis. In Proceedings of the 2024 IEEE/ACM 46th international conference on software engineering: Companion proceedings, pp. 364-365, 2024.*
>
> *[2] Liu, Y., Ji, Y., Tao, S., He, M., Meng, W., Zhang, S., Sun, Y., Xie, Y., Chen, B., Yang, H. Loglm: From task-based to instruction-based automated log analysis. In 2025 IEEE/ACM 47th International Conference on Software Engineering: Software Engineering in Practice (ICSE-SEIP), pp. 401-412, 2025.*
>
> *[3] Zhou, C., Liu, Y., Meng, W., Tao, S., Tian, W., Yao, F., Li, X., Han, T., Chen, B., Yang, H. Srdc: Semantics-based ransomware detection and classification with llm-assisted pre-training. In Proceedings of the AAAI Conference on Artificial Intelligence, pp. 28566-28574, 2025.*

---

> > ### Author Rebuttal · Reviewer_4rDu · 2026-04-01
> >
> > Thanks to the authors for the detailed rebuttal. My main concerns are adequately addressed.

---

> > > ### Author Response · Authors · 2026-04-06
> > >
> > > Thank you very much for your careful review and for taking the time to revisit the paper after our rebuttal. We’re really glad to hear that your concerns have been addressed.
> > >
> > > We truly appreciate your thoughtful feedback—it has been very helpful for us in clarifying and strengthening the work. If you feel it is appropriate, we would be grateful if the score could reflect your current assessment.
> > >
> > > Thanks again for your time and support!

---

### Official Review · Reviewer_1Anv · 2026-03-13

**Soundness:** 3
**Presentation:** 3
**Significance:** 4
**Originality:** 4
**Overall Recommendation:** 5
**Confidence:** 3

**Summary:**

This paper presents a novel method to reason on ultra-long logs for cybersecurity applications. The logs are processed as though they are visual embeddings in a hierarchical manner with necessary token compression imbibed through cross-attention and learnt queries. The learning process is three-stage similar to some of the latest VLM works. First, embeddings from the logs are organized as images, sent through VIT architecture and are aligned with corresponding text summaries from a GPT model. Several of these "image" tokens are further compressed through "spatiotemporal" cross-attention layers and finally aligned with an LLM. Experimental results show huge gains over existing baselines, both text-only and domain-specific.

**Compliance With Llm Reviewing Policy:**

Affirmed.

**Final Justification:**

Rebuttal has addressed my concerns and I retain my recommendation of Accept.

**Key Questions For Authors:**

Are the vision and language encoders initialized with existing pre-trained models?

**Limitations:**

yes

**Strengths And Weaknesses:**

## Strengths:

- The paper is well-written. It introduces a novel parallel between logs and vision domain and applies SOTA vision-language model techniques, starting from image-text alignment followed by video-text alignment and finally video-LLM fine-tuning to the domain of logs.
- It's illuminating to see that the logs can be hierarchically compressed similar to vision tokens, perhaps because of a similar sparsity of information between logs and images. The formulation looks thorough.
- In-domain (Table2) and out-of-domain tests (Table3, 4) both show that the proposal achieves SOTA beating both long-context LLM baselines that read the entire logs and also the past architectural choices.
- Table 5 clearly shows the huge efficiency gain over just passing the logs to a transformer.
- Table 7 demonstrates the positive scaling behavior

## Weaknesses:
- The paper doesn't discuss much in the literature survey about the video encoder works that it seems to be heavily inspired from [1, 2, 3, 4]
- The paper doesn't talk much about "why" such an interesting parallel exists between logs and VLMs. Understanding this parallel can help the readers expand the scope of this work to other domains.
- A little more introduction to the logs and related cybersecurity problems can help readers from the vision domain get a primer.

[1] VideoCoCa: Video-Text Modeling with Zero-Shot Transfer from Contrastive Captioners \
[2] VideoPrism: A Foundational Visual Encoder for Video Understanding \
[3] InternVideo2: Scaling Foundation Models for Multimodal Video Understanding \
[4] LLaVA-Video: Video Instruction Tuning With Synthetic Data

---

> ### Author Rebuttal · Authors · 2026-03-31
>
> We sincerely thank the reviewer for recognizing the contributions of our work and for the encouraging recommendation. We have carefully considered all comments and will incorporate corresponding improvements in the revised manuscript.
>
> **R1: Related work on video encoders**
>
> We thank the reviewer for pointing this out. We will add a dedicated subsection (section 2.5) in the related work to discuss representative video-text models, including VideoCoCa, VideoPrism, InternVideo2, and LLaVA-Video.
>
> **R2: Why logs and VLMs share a parallel**
>
> We agree this is an important point. We will add a dedicated explanation in a newly introduced Section 2.5 of the revised manuscript to explicitly explain why such a parallel between logs and VLMs exists.
>
> Specifically, both modalities exhibit high redundancy with sparse salient signals, requiring the model to identify and align semantically important elements across space and time. Under this view, EDR logs can be naturally interpreted as a spatio-temporal structure, where fields within an event form “spatial” elements and event sequences capture temporal dynamics. This perspective motivates the use of VLM-style designs (image → video → language alignment) for EDR logs, and highlights their applicability to other structured sequential data.
>
> **R3: Primer for readers from the vision/NLP community**
>
> We thank the reviewer for this helpful suggestion. We will add a new section in the Appendix of the revised manuscript to provide a brief introduction to EDR logs, covering: (i) What EDR logs are, (ii) Typical event structures (process, file, network interactions) and (iii) why ultra-long logs with sparse malicious signals pose unique challenges. This will help readers from the vision or NLP communities better understand the problem context and significance.
>
> **R4: Model initialization**
>
> Yes. The vision transformer and language model are initialized from standard pre-trained checkpoints, while the kvEmbedding and cross-attention modules are trained from scratch. This typically provides a strong initialization for the model.

---

> > ### Author Rebuttal · Reviewer_1Anv · 2026-04-04
> >
> > Thank you authors.

---

> > > ### Author Response · Authors · 2026-04-06
> > >
> > > We sincerely appreciate your thorough review and valuable suggestions. We are pleased to know that our responses have resolved your concerns, and we thank you for helping us improve the manuscript.

---

### Official Review · Reviewer_meJv · 2026-03-13

**Soundness:** 2
**Presentation:** 2
**Significance:** 3
**Originality:** 3
**Overall Recommendation:** 4
**Confidence:** 3

**Summary:**

This paper introduces WatchLog, a multimodal framework for malicious behavior detection from Endpoint Detection and Response (EDR) logs. The main idea is quite interesting: it turns log events into image-like representations, where key-value fields are mapped into pixel-level embeddings through a learned kvEmbedding module, and then organizes these event images as a temporal video sequence. On top of this, the authors design a three-stage training pipeline, including image-event alignment pretraining, video-log alignment for temporal compression, and supervised fine-tuning for both threat family classification and rationale generation.

The paper also prresents EDR8M-20R, a large-scale dataset covering 8 million events and 20 behavior families with expert annotated rationales, which is a valuable contribution by itself. Experimental results are strong, with high detection accuracy and much better efficiency than LLM-based baselines, especially for very long logs. The methods also shows decent transfer ability on out-of-distribution attacks and the ATLASv2 benchmark. Ablation results are also fairly complete, covering training stages, compression settings, and model scale. Overall, the paper is well-motivated and technically solid, though some parts of the presentation feels a bit dense.

**Compliance With Llm Reviewing Policy:**

Affirmed.

**Final Justification:**

As mentioned in the Summary & Strengths And Weaknesses, generally the paper has more good inspiration than some minor points to be revised. I will say it's a weak acc.

**Key Questions For Authors:**

* why the video formulation is really needed over simpler compression methods. The paper does not clearly show the empirical benefit of using image/video representations instead of more direct sequence compression, such as chunk-level summarization, hierarchical attention, or even simple pooling on token embeddings. An ablation with a purely text-based compression baseline would be very useful here, and honestly could change my view of the paper’s main contribution.
* I also have concerns about statistical reliability. The test set only has 20 samples per family, so reported differences may not be very stable. It would help a lot to report confidence intervals, or variance across multiple runs. Right now, even one or two samples can shift the results non-trivially.
* Another issue is the role of spatial structure. Table 14 suggests the model is fairly robust to patch permutation, which kind of weakens the claim that spatial layout is important. If spatial arrangement is not doing much, then why not just process a flat kvEmbedding sequence?

**Limitations:**

yes

**Strengths And Weaknesses:**

* The paper studies a very practical problem: how to analyze extremely long EDR logs efficiently. I think the main idea is quite creative, namely converting structured logs into video-style representations and then leveraging multimodal models on top. The efficiency improvement is also pretty impressive. For example, handling 1M tokens with around 49GB memory and about 1 second latency, compared with 266GB and 522s for LLaMA3.1-8B, is a strong result. The 3-stage training pipeline is well designed and the ablation section is fairly thorough, showing each component matters. Also, the EDR8M-20R dataset seems like a useful contribution, since there is indeed not many public resources of this kind. The appendix math analysis also helps, at least partially, to justify the design.

* The evaluation set feels rather small, only 20 test samples per family, so the statistical reliability is a bit questionable. The ATLASv2 result is not especially strong either, which may indicate the generalization is still limited.
* For rationale quality, the paper depends fully on LLM-as-a-judge, and this is somewhat concerning because the rationales themselves are also LLM-generated.
* the log-to-video angle is new, many building blocks are quite standard, and the paper does not fully convince me why this representation is better than other compression methods.

---

> ### Author Rebuttal · Authors · 2026-03-31
>
> Thank you for the insightful and constructive questions. We address each point below with additional clarifications and empirical evidence.
>
> **R1: Why is the video formulation necessary compared to simpler compression methods?**
>
> We appreciate this fundamental question. Our key observation is that **EDR logs exhibit an inherent spatio-temporal structure**: **(i) within each event**, key-value fields encode structured semantics that can be naturally organized as “spatial” pixels, and **(ii) across events**, malicious behaviors evolve over time with strong temporal dependencies. The video formulation explicitly models both aspects, enabling joint modeling of intra-event semantics and inter-event dynamics, which is difficult for flat sequence compression to preserve.
>
> To further validate this, **we conducted additional experiments with purely text-based compression baselines**, including pooling-based compression (event-level), hierarchical attention (intra-event→inter-event causal attention), and event-level summarization. For a fair comparison, all methods use an LLM of the same size as that used in WatchLog, with efficiency metrics are evaluated under the same Transformers library and hardware setup on 1024K log token inputs.
>
> | | BA↑ | R↑ | FA↓ | OA↑ | Cohe.↑ | Cons.↑ | Comp.↑ | GPU-MU (GB) | TTFT (s) |
> |:-|:-|:-|:-|:-|:-|:-|:-|:-|:-|
> | Qwen2.5-1.5B-Pool | 95.0 | 100.0 | 100.0 | 22.8 | 93.0 | 93.8 | 40.7 | **6.36** | **0.04** |
> | Qwen2.5-1.5B-HieAttention | 98.0 | 99.5 | 30.0 | 3.8 | 92.9 | 95.0 | 28.8 | 39.42 | 0.44 |
> | Qwen2.5-1.5B-Summary | 98.5 | 98.4 | **0.0** | 85.5 | 93.1 | 93.1 | **80.3** | 21.51* | 13.19* |
> | WatchLog | **99.8** | **100.0** | 5.0 | **90.3** | 93.6 | **95.0** | 68.5 | 48.61 | 1.11 |
>
> As shown in the table, pooling-based compression and hierarchical attention improve inference efficiency but **lead to noticeable degradation in both detection and reasoning quality**. Event-level summarization reduces input length by ~80% compared to raw logs and achieve strong performance in reasoning quality, yet **still lags behind WatchLog in detection metrics and inference time. Moreover, it requires an additional high-quality summarization model during inference, introducing extra latency and increased system complexity**. Notably, results marked with * exclude the summarization overhead, thus underestimating the true end-to-end cost.
>
> These results indicate that purely text-based compression methods struggle to preserve fine-grained behavioral semantics in long-context scenarios. In contrast, WatchLog achieves superior performance not merely due to compression, but due to its event-level semantic alignment with explicitly modeled temporal dependencies, which is essential for capturing complex EDR log behaviors.
>
> **R2: Statistical reliability of results**
>
> We fully agree that statistical robustness is important. To this end, **we conducted five independent runs** with different random seeds and report results as mean ± standard deviation. The results show consistently low variance (e.g., 0.78 and 1.1 for OA and Comp., respectively), indicating stable performance. We will include these results in the revised manuscript.
>
> | | BA↑ | R↑ | FA↓ | OA↑ | Cohe.↑ | Cons.↑ | Comp.↑ |
> |:-|:-|:-|:-|:-|:-|:-|:-|
> | WatchLog | 99.7±0.2 | 99.8±0.2 | 3.0±2.4 | 89.4±0.78 | 90.9±1.4 | 92.5±1.5 | 66.6±1.1 |
>
> In addition, we would like to clarify (apologies for the lack of emphasis) that the paper already **includes an Independent Test Set (Lines 254–256 and Table 3), consisting of ~1000 newly collected samples**. The results on this larger set are consistent with those in Table 2, further supporting the reliability of our findings.
>
> **R3: Role of spatial structure under patch permutation**
>
> We thank the reviewer for this insightful observation. We would like to clarify that the robustness to patch permutation shown in Table 14 **does not imply that spatial structure is unimportant; rather, it indicates that the model is invariant to absolute spatial positions**.
>
> The key distinction is that our model does not rely on fixed positional layouts, but still benefits from a structured 2D organization of fields. Specifically, the 2D layout provides a consistent structural prior that **supports the subsequent pixel-level cross-attention in the Log-to-Video module, helping preserve field-level alignment across events as well as temporal dependencies**. This enables the model to maintain event semantic consistency during both training and inference, even under structural perturbations such as patch permutation.
>
> In contrast, directly flattening kvEmbedding outputs into a 1D sequence disrupts this alignment mechanism, weakening temporal correspondence and degrading semantic consistency. To further verify this, **we flattened kvEmbedding outputs and evaluated them under the same setting**. The resulting performance is substantially lower, with FA and OA reaching only 15.0% and 73.8%, respectively—far below our method.

---

> > ### Author Rebuttal · Reviewer_meJv · 2026-04-06
> >
> > Thank the authors for the detailed and thoughtful rebuttal, I will maintain my original score.

---

### Decision · Program_Chairs · 2026-04-30

**Decision:**

Accept (regular)

**Comment:**

This paper received one accept, two weak accept, and one weak reject as the final rating. While reviewers overall agree with the technical contribution and feel the paper is well structured and presented, there are questions regarding the experiments, motivation of the method, and comparisons with existing approaches. The rebuttal addresses some of the major concerns, but the weak reject reviewer still has remained questions.  After reading the paper, rebuttal, and reviewer comments, AC agrees weak accept.